

**Multi-gas and multi-source comparisons of six land use emission datasets**
**and AFOLU estimates in the Fifth Assessment Report**
**Short title:** AFOLU dataset comparisons
**Authors affiliation:**
Rosa Maria Roman-Cuesta[1,2]*, Martin Herold[2], Mariana C. Rufino[1], Todd S. Rosenstock[3],
Richard A. Houghton[4], Simone Rossi[5], Klaus Butterbach-Bahl[6,7], Stephen Ogle[8], Benjamin
Poulter[9], Louis Verchot[10,11], Christopher Martius[1].
[1] Center for International Forestry Research (CIFOR), P.O Box 0113 BOCBD, Bogor 16000,
Indonesia.
[2] Laboratory of Geo-Information Science and Remote Sensing - Wageningen University.
Droevendaalsesteeg 3, 6708PB. Wageningen. The Netherlands.
[3] World Agroforestry Centre (ICRAF). PO Box 30677-00100, Nairobi. Kenya.
[4] Woods Hole Reseach Center. 149 Woods Hole Road Falmouth, MA, 02540-1644, US.
[5] Global Environmental Monitoring Unit, Institute for Environment and Sustainability,
European Commission, Joint Research Centre, TP. 440 21020 Ispra, Varese 21027, Italy,
[6] International Livestock Research Institute (ILRI) P.O. Box 30709. Nairobi 00100, Kenya
[7] Karlsruhe Institute of Technology, Institute of Meteorology and Climate Research (IMK-
IFU), Garmisch-Partenkirchen, Germany
[8] Natural Resource Ecology Laboratory, Campus Delivery 1499, Colorado State University,
Fort Collins, Colorado 80523-1499, USA.





[9] Ecosystem Dynamics Laboratory. Montana State University. P.O. Box 172000. Bozeman,
MT 59717-2000. USA.
[10] International Center for Tropical Agriculture, Km17 Recta Cali-Palmira, Apartado Aéreo
6713, Cali, Colombia.
[11] Earth Institute Center for Environmental Sustainability, Columbia University, New York,
USA.
* Corresponding author. Telephone:+31317485919, Fax:  Email: rosa.roman@wur.nl
**Keywords**: AFOLU, Land use greenhouse gas emissions, Land Use Land Cover Change and
Forestry, LULUCF, mitigation, Fifth Assessment Report, gross emissions flux.

## ABSTRACT

The Agriculture, Forestry and Other Land Use (AFOLU) sector contributes with ca. 20-25%
of global anthropogenic emissions (2010), making it a key component of any climate change
mitigation strategy. AFOLU estimates remain, however, highly uncertain, jeopardizing the
mitigation effectiveness of this sector. Global comparisons of AFOLU emissions have shown
divergences of up to 25%, urging for improved understanding on the reasons behind these
differences. Here we compare a diversity of AFOLU emission datasets (e.g. FAOSTAT,
EDGAR, the newly developed AFOLU "Hotspots", "Houghton", "Baccini", and EPA) and
estimates given in the Fifth Assessment Report, for the tropics (2000-2005), to identify
plausible explanations for the differences in: i) aggregated gross AFOLU emissions, and ii)
disaggregated emissions by sources, and by gases ($CO_2$, $CH_4$, $N_2O$). We also aim to iii)
identify countries with low agreement among AFOLU datasets, to navigate research efforts.



Aggregated gross emissions were similar for all databases for the AFOLU: 8.2 (5.5-12.2), 8.4
and 8.0 Pg $CO_2e.yr^{-1}$ (Hotspots, FAOSTAT and EDGAR respectively), Forests: 6.0 (3.8-10),
5.9, 5.9 and 5.4 $PgCO_2e.yr^{-1}$ (Hotspots, FAOSTAT, EDGAR, and Houghton), and
Agricultural sectors: 1.9 (1.5-2.5), 2.0, 2.1, and 2.0 $PgCO_2e.yr^{-1}$ (Hotspots, FAOSTAT,
EDGAR, and EPA). However, this agreement was lost when disaggregating by sources,
continents, and gases, particularly for the forest sector (fire leading the differences).
Agricultural emissions were more homogeneous, especially livestock, while croplands were
the most diverse. $CO_2$ showed the largest differences among datasets. Cropland soils and
enteric fermentation led the smaller $N_2O$ and $CH_4$ differences. Disagreements are explained
by differences in conceptual frameworks (e.g. carbon-only *vs* multi-gas assessments,
definitions, land use versus land cover, etc), in methods (Tiers, scales, compliance with
Intergovernmental Panel on Climate Change (IPCC) guidelines, legacies, etc) and in
assumptions (e.g. carbon neutrality of certain emissions, instantaneous emissions release, etc)
that call for more complete and transparent documentation for all the available datasets.
Enhanced dialogue between the carbon ($CO_2$) and the AFOLU (multi-gas) communities is
needed to reduce discrepancies of land use estimates.

## 1. INTRODUCTION

Modelling studies suggest that to keep the global mean temperature increase to less than 2°C
and to remain under 450 ppm of $CO_2$ by 2100, $CO_2$ emissions must be cut 41-72% below
2010 levels by 2050 (IPCC, 2014), and global emissions levels must be reduced to zero (a
balance between sources and sinks) before 2070 and below zero, through removal processes,
after that (Anderson, 2015; UNEP, 2015). To reach these ambitious goals, tremendously rapid
improvements in energy efficiency and nearly a quadrupling of the share of zero and low
carbon energy supply (e.g. renewables, nuclear energy, and carbon dioxide capture and



storage (CCS), including bioenergy (BECCS)) would be needed by 2050 (IPCC, 2014;
Friedlingstein et al., 2014; Anderson, 2015; UNEP, 2015). Since there is no scientific
evidence on the feasibility of CCS technologies (Anderson, 2015), renewables and the land
use sector are among the most plausible options (Canadell and Schulze, 2014). Optimistic
estimates suggest that the AFOLU sector (here indistinctively also called land use sector)
could contribute from 20 to 60% of the total cumulative abatement to 2030 including
bioenergy (Smith et al., 2014).

The AFOLU sector roughly contributes with a quarter (10-12 $PgCO_2e.yr^{-1}$) of the total
anthropogenic GHG emissions (50 $PgCO2e.yr^{-1}$) (Smith et al., 2014) through a few human
activities: deforestation, forest degradation, and agriculture including cropland soils, paddy
rice, and livestock (Smith et al., 2014). Despite the acknowledged importance of the
emissions from the land use sector in global mitigation strategies, assessing GHG emissions
and removals from this sector remains technically and conceptually challenging (Abad-Viñas
et al., 2014; Ciais et al., 2014). This challenge relates to an incomplete understanding of the
processes that control the emissions from the land use sector (Houghton et al., 2012),
especially post-disturbance dynamics (Frank et al., 2015; Poorter et al., 2016) and to various
sources of error that range from inconsistent definitions, methods, and technical capacities
(Romijn et al., 2012, 2015; Abad-Viñas et al., 2014), to special features of the land use sector
such as legacy and reversibility/non-permanence effects (Estrada et al., 2014), or to the
difficulty to separate anthropogenic from natural emissions (Estrada et al., 2014; Smith et al.,
2014). As a result, the AFOLU emissions are the most uncertain of the all the sectors in the
global budget, reaching up to 50 percent of the emissions mean (Houghton et al., 2012; Smith
et al., 2014; Tubiello et al., 2015). This is important since uncertainties jeopardize the
effectiveness of the AFOLU sector to contribute to climate change mitigation. Thus, country



compliances to their mitigation targets are likely to be controversial when the uncertainty is
equal to or greater than the pledged emission reductions (Grassi et al., 2008; Pelletier et al.,

99   2015).


Currently, data on AFOLU emissions are available through national greenhouse gas
inventories, which are submitted to the United Nations Framework Convention on Climate
Change (UNFCCC), but these national estimates cannot be objectively compared due to
differences in definitions, methods, and data completeness (Houghton et al., 2012; Abad-
Viñas et al., 2014). More comparable AFOLU data are offered in global emission databases
such as EDGAR or FAOSTAT (Smith et al., 2014; Tubiello et al., 2015), or more sectorial
datasets such as the Houghton's Forestry and other Land Use (FOLU) data (Houghton et al.,
2012), and the US Environmental Protection Agency non-$CO_2$ emissions for agriculture -
including livestock (USEPA, 2013). While national inventories and global databases are
currently the best bottom up emissions data we count on, their utility to inform on what the
atmosphere receives has been contested. Late research shows disagreements between the
trends of reported emissions and atmospheric growth since 1990 for $CO_2$ (Francey et al.,
2010, 2013a, 2013b), for $CH_4$ (Montzka et al., 2011), and for $N_2O$ (Francey et al., 2013b). In
the case of $CO_2$, Francey *et al.* conclude that the differences between atmospheric and
emission trends for $CO_2$ might be more related to under-reported emissions (~9 PgC for the
period 1994-2005), than to adjustments in the terrestrial sinks (e.g. increased $CO_2$ removals in
oceans and forests). On the other hand, global AFOLU databases suffer from inconsistencies
that  lead to global $CO_2$e emissions differences of up to 25% (2000-2009) (Tubiello et al.,
2015): 12.7 vs 9.9 $PgCO_2e.yr^{-1}$ for EDGAR and FAOSTAT, respectively. These datasets also
disagreed in the contribution of the AFOLU sector to the total anthropogenic budget in 2010
(e.g. 21% and 24% for FAOSTAT vs EDGAR), and on the relative share of the emissions



from agriculture versus FOLU since 2010. Thus, while EDGAR implies a relatively equal
contribution (IPCC, 2014), FAOSTAT reports agricultural emissions being larger contributors
to the total anthropogenic budget ($11.2\pm0.4\%$) than forestry and other land uses ($10\pm1.2\%$)
(Tubiello et al., 2015), with a steady growth trend of 1% since 2010.

Understanding the inconsistencies among AFOLU datasets is an urgent task since they
preclude our accurate understanding of land-atmosphere interactions, GHG effects on climate
forcing and, consequently, the utility of modelling exercises and policies to mitigate climate
change (Houghton et al., 2012; Grace et al., 2014; Smith et al., 2014; Sitch et al., 2015; Tian
et al., 2016). The land use sector plays a prominent role in the Paris Agreement (Art.5), with
many countries including it as mitigation targets in their Nationally Determined Contributions
(NDCs) (Grassi and Dentener, 2015; Richards et al., 2015; Streck, 2015). It is then urgent to
understand how much and why different AFOLU datasets differ in their emissions estimates,
so that we can better navigate countries' land-based mitigation efforts, and help to validate
their proposed claims under the UNFCCC.

Here we compare gross AFOLU emissions estimates for the tropics, for 2000-2005, from six
datasets: FAOSTAT, EDGAR, "Houghton", "Baccini", the US Environmental Protection
Agency data (EPA), and a recently produced, spatially explicit AFOLU dataset, that we will
hereon call "Hotspots" (Roman-Cuesta *et al,. under review).*
). We aim to identify differences and plausible explanations behind: i) aggregated AFOLU,
FOLU and Agricultural gross emissions, ii) disaggregated contributions of the emission
sources for the different datasets, iii) disaggregated contribution of the different gases ($CO_2$,
$CH_4$, $N_2O$), and iv) national scale disagreements among datasets.



**2. METHODS**
*2.1 Study area*
Our study area covers the tropics and the subtropics, including the more temperate regions of
South America (33° N to 54° S, 161° E to 117° W). Land use change occurs nowhere more
rapidly than in this region (Poorter et al., 2016), so its study has global importance. We
selected the period 2000-2005 for being the common temporal range for all the datasets. This
period is not for the recent past but that does not affect the comparative nature of this
research. Our study area focuses at the country level and includes eighty countries, following
Harris et al., (2012). We ran the comparisons on gross emissions. Mitigation action can be
directed to reducing emissions by the sources, or to increasing the absorptions by the sinks, or
to both. While gross and net emissions are equally important, they offer different information
(Richter and Houghton, 2011; Houghton et al., 2012). Net land use emissions consider the
emissions by the sources and the removals by the sinks in a final emission balance where the
removals are discounted from the emissions, closer to what the atmosphere receives. Land use
sinks refer to any process that stores GHGs (e.g. forest growth, forest regrowth after
disturbances, organic matter stored in soils, etc) (Richter and Houghton, 2011). Gross
assessments can consider both the emissions produced by the sources (gross emissions) and
the removals absorbed by the sinks (gross removals), but they are not offered in a final
balance where the sinks are discounted from the sources. They are offered as separate fluxes,
instead. They are useful to navigate mitigation implementation since they offer direct
information on the sources and sinks that need to be acted upon through policies and
measures to enhance and promote mitigation. However, lack of ground data makes the
assessment of the sinks much more difficult than the assessment of the sources (Houghton et
al., 2012; Grace et al., 2014; Brienen et al., 2015) with a particular gap on disturbed standing
forests (Poorter et al., 2016). For these reasons, we here focus on gross emissions by the



sources, excluding gross sinks.

***2.2 AFOLU datasets***
*Hotspots:* this is a multi-gas ($CO_2$, $CH_4$, $N_2O$) spatially explicit (0.5°) database on gross
AFOLU emissions and associated uncertainties for the tropics for the period 2000-2005, at
Tier 2 and Tier 3 levels. It identifies Hotspots of AFOLU emissions to help prioritize
mitigation actions. It combines available published GHG datasets for the key sources of
emissions in the AFOLU sector, as identified by the Fifth Assessment Report (AR5) of the
Intergovernmental Panel on Climate Change (Smith et al., 2014): deforestation, forest
degradation (fire, wood harvesting), crop soils, paddy rice, and livestock (enteric fermentation
and manure management). Tier 1 emission estimates of agricultural peatland decomposition
are also included. Forest emissions mainly report aboveground biomass (except fire that also
reports on soils). More detailed methodological information is available in Roman-Cuesta et
al., (under review).

*FAOSTAT:* covers agriculture, forestry and other land uses and their associated emissions of
$CO_2$, $CH_4$ and $N_2O$, following IPCC, 2006 Guidelines at Tier 1 (Tubiello et al., 2013, 2014).
Emissions are estimated for nearly 200 countries, annually, for the reference period of 1961–
2012 (agriculture) and 1990–2012 (FOLU), based on national activity data submitted by
countries and further collated by FAO. Projected emission data are available for 2030 and
2050. FAOSTAT includes estimates of emissions from biomass fires, peatland drainage and
fires, based on geo-spatial information, as well as on forest carbon stock changes (both
emissions and removals) based on national-level FAO Forest Resources Assessment data
(FRA 2010).



*EDGAR:* The Emissions Database for Global Atmospheric Research (EDGAR) provides
global GHG emissions from multiple gases ($CO_2$, $CH_4$, $N_2O$, HFCs, PFCs and SF6) at 0.1°
and country levels. The EDGAR database covers all IPCC sectors (energy, industry, waste
management, and AFOLU), mostly applying IPCC 2006 guidelines for emission estimations
(EDGAR 2012). We downloaded the EDGAR's 4.2 Fast Track 2010 (FT 2010). FT 2010
emissions cover the period 2000-2010 in an annual basis, at the country level.

*"Houghton":* Houghton's bookkeeping model calculates the net and gross fluxes of carbon
($CO_2$ only) between land and atmosphere that result from land management (Houghton, 1999,
2012; Houghton and Hackler, 2001; Houghton et al., 2012). The net estimate includes
emissions of $CO_2$ from deforestation, shifting cultivation, wood harvesting, wood debris
decay, biomass burning (for deforestation fires only, peatland fires were not included in our
version of their data), and soil organic matter from cultivated soils. It also includes sinks of
carbon in forests recovering from harvest and agricultural abandonment under shifting
cultivation. Unlike the other datasets, all pools are included: live vegetation, soil, slash
(woody debris produced during disturbance), and wood products. The model does, however,
not include forests that are not logged, cleared or cultivated. Rates of growth and
decomposition are ecosystem specific and do not vary in response to changes in climate, $CO_2$
concentrations, or other elements of environmental change. Therefore, forests grow (and
wood decays) at the same rates in 1850 and 2015. Unlike other databases all carbon in the
ecosystem considered is accounted for: live vegetation, soil, slash (woody debris produced
during disturbance), and wood products. We downloaded regional annual emissions from the
TRENDS (1850-2005) dataset for the tropics: Central and South (CS) America, tropical
Africa and South and South East Asia. Only net emissions were available. No spatially
disaggregated data were offered (e.g. countries). Houghton's data are, unlike all the other



datasets, net aggregated FOLU estimates, for $CO_2$-only.

*"Baccini":* These are gross FOLU tropical emissions that derive from Houghton's
bookkeeping model and published by Baccini et al., (2012). Data are gross disaggregated
emissions estimates for the period 2000-2010: deforestation (4.18 $PgCO_2.yr^{-1}$), wood
harvesting (1.69 $PgCO_2.yr^{-1}$), biomass burning (2.86 $PgCO_2.yr^{-1}$), wood debris decay (3.04
$PgCO_2.yr^{-1}$). Baccini's estimates refer, however, to a tropical area slightly smaller than our
study region.

*The US Environmental Protection Agency (EPA):* global non-$CO_2$ projected emissions for the
period 1990-2030 for the Agriculture, Energy, Industrial Processes and Waste sectors, for
more than twenty gases. EPA uses future net emissions projections of non-$CO_2$ GHGs as a
basis for understanding how future policy and short-term, cost-effective mitigation options
can affect these emissions. EPA follows the Global Emissions Report, which uses a
combination of country-prepared, publicly-available reports consistent with IPCC guidelines
and guidance (USEPA, 2013). When national emissions estimates were unavailable, EPA
produced its own non-$CO_2$ emissions using IPCC methodologies (e.g., international statistics
for activity data, and the default IPCC Tier 1 emission factors). Deviations to this
methodology are discussed in each of the source-specific methodology sections of USEPA
(2012). No FOLU estimates are included in this dataset. We downloaded agricultural
emissions offered as 5-year intervals at country level, disaggregated by gas ($N_2O$ and $CH_4$),
and by emission sources.

*IPCC AR5*: The AR5 is a synthesis report, not a repository of global data. However, new
AFOLU data are produced by the merging of peer-reviewed data such as Figures 11.2, 11.4,



11.5 and 11.8 in chapter 11 of the AR5 (Smith et al., 2014). We will contrast our six datasets
against the data from these newly produced figures.

Table 1 shows a summary of key similarities and differences of the assessed AFOLU datasets
and the data from the AR5. The exact variables used for each database, are described in Table
S1 in the supplementary material (SI). Datasets can be downloaded at the websites described
in the reference section.

*2.3 Estimating comparable gross AFOLU emissions for all datasets*
We focus on human-induced gross emissions only, excluding fluxes from unmanaged land
(e.g. natural wetlands). We focus on direct emissions excluding indirect emissions whenever
possible (e.g. nitrate leaching and surface runoff from croplands). Delayed fluxes (legacies)
are important (e.g. underestimations of up to 62% of the total emissions when recent legacy
fluxes are excluded) (Houghton et al., 2012) but are frequently omitted in GHG assessments
that derive from remote sensing, such as some of the datasets used in this comparison (e.g.
deforestation emissions from Harris *et al.* (2012)). Wood harvesting emissions also excluded
legacy fluxes. We assumed instantaneous emissions of all carbon that is lost from the land
after human action (Tier 1, IPCC 2006) (e.g. deforested and harvested wood), with no
transboundary considerations (e.g. the emissions are assigned wherever the disturbance takes
place, particularly important for  Harvested Wood Products). Life-cycle substitution effects
were neither considered for harvested wood (Peters et al., 2012). Some exceptions were
allowed when data were already aggregated (e.g. for Houghton's and EPA's datasets we could
not exclude indirect emissions linked to forest decay and agriculture, respectively), or because
their legacy (past decay) estimates corresponded to an important source (e.g. EDGAR's post
burned decay and decomposition emissions represent deforestation) (Tubiello et al., 2015). To



facilitate comparisons, emissions estimates included the exact same emission sources:
deforestation, wood harvesting, fire, livestock (enteric fermentation + manure management),
cropland soil emissions, rice emissions, emissions from drained histosols), for $CO_2$, $CH_4$, and
$N_2O$. See Table S1 in SI to review the exact sources used in each database. Fire emissions do
not include $CO_2$ emissions from biomass burning in non-woody vegetation -savannas and
agriculture – as they are assumed in equilibrium with annual regrowth processes (for $CO_2$
gases only) (IPCC 2003, 2006).

### 2.4 Correcting known differences among datasets estimates

Tubiello *et al.* (2015) identified four main differences that resulted in larger estimates for the
EDGAR data than for FAOSTAT, under the AFOLU estimates of the AR5 (Smith et al.,
2014): 1. The inclusion of energy emissions under the agriculture budget, 2. Inclusion of
savannah burning, 3. Higher rice emissions due to the use of the IPCC 1996 guidelines instead
of the IPCC 2006 guidance, 4. FOLU's unresolved differences due to unclear metadata on
EDGAR's proxy for deforestation (post burned decay and decomposition). We have corrected
for the first two in our data comparison. No energy, and no $CO_2$ for savannah burning have
been included in the AFOLU estimates in any of our analyses.

### 2.5 Country emissions

We estimated the country-scale level of emissions agreement for the three most complete
databases: FAOSTAT, EDGAR and Hotspots using the coefficient of variation among data,
for AFOLU, forests (deforestation, fire and wood harvesting), crops (cropland soils, paddy
rice) and livestock emissions. Percentiles were then used to separate between countries with
high level of agreement (≥75[th] percentile), moderate agreement (50[th]-75[th]), low agreement
(25[th]-50[th]), and very low agreement (≤25[th]).




## 3. RESULTS AND DISCUSSION

### 3.1 Aggregated AFOLU, FOLU and Agricultural emissions

We found good agreement among datasets for the aggregated tropical scales with AFOLU values of 8.0 (5.5-12.2) (5th-95th percentiles), 8.4 and 8.0 $PgCO_2e.yr^{-1}$ (for the Hotspots, FAOSTAT and EDGAR, respectively). FOLU (deforestation and forest degradation) contributed with 6.0 (3.8-10), 5.9, 5.9 and 5.4 $PgCO_2e.yr^{-1}$ for the Hotspots, FAOSTAT, EDGAR, and Houghton datasets respectively. Agriculture (livestock, cropland soils and rice emissions) reached 1.9 (1.5-2.5), 2.5, 2.1, and 2.0 $PgCO_2e.yr^{-1}$ for the Hotspots, FAOSTAT, EDGAR, and EPA datasets respectively (Figure 1, Table 2). Forest emissions represented ≥70% of the tropical AFOLU gross mean annual budget for 2000-2005 (our Hotspots database and Houghton showing the highest and the lowest estimates), and agriculture represented the remaining 25-30% AFOLU emissions (FAOSTAT and Hotspots showing the highest and the lowest values). Houghton's FOLU value (5.4 $PgCO_2.yr^{-1}$) is a net estimate that includes carbon dynamics associated to forest land use changes, and forest removals from areas under logging and shifting cultivation and it is, as expected, lower than the forest gross emissions. Its value for the tropics was, however, higher than the net FOLU value used in the IPCC AR5 (4.03 $PgCO_2e.yr^{-1}$ for 2000-2009) (Houghton *et al.* 2012). Since boreal and temperate forest sinks are reported to be quasi-neutral (Houghton et al., 2012), these differences are unclear. There is a variety of Houghton' net FOLU estimates in current bibliography (e.g. 4.03 $PgCO_2e.yr^{-1}$ for 2000-2009 in Smith *et al.* (2012), 4.9 for 2000 and 4.2 for 2010 (Tubiello et al., 2015) that likely correspond to different updates of the same dataset, but create confusion and would call for verified official values that could be consistently used.






The IPCC AR5 offers a FOLU gross value for the tropics of ca. 8.4 $PgCO_2.yr^{-1}$ (2000-2007)
(Fig 11.8 in AR5, Smith et al., 2014) (Fig S1, SI) which corresponds to Baccini's estimates
using Houghton's bookkeeping model. This value is in the upper range of our gross FOLU
emissions: 6 (3.8-10) $PgCO_2e.yr^{-1}$ (2000-2005), and higher than the mean gross FOLU
emissions from all the other datasets (approx. 6 $PgCO_2e.yr^{-1}$) (Table 2). The time periods are
not identical and we do not compare the same gases (e.g. the bookkeeping model focuses on
$CO_2$ only, while we run a multi-gas assessment). However, the differences mainly relate to
unreported choices behind the inclusion/exclusion of emission sources and the description of
their methods, in the AR5. Thus, the 8.4 $PgCO_2.yr^{-1}$ gross estimate does not include fire, and
has larger contributions from shifting cultivation (2.35 $PgCO_2.yr^{-1}$) and wood-harvesting (2.49
$PgCO_2.yr^{-1}$), than the deforestation and wood-harvesting emissions in our selected datasets
(Figure 2). Numbers used in Figure 11.8 also exclude other gross emissions offered in Baccini
*et al.* (2012), which is the citation used in Fig. 11.8. Explicit, complete, and transparent
documentation is encouraged for the next AFOLU figures in the IPCC Assessment Reports.
Another consideration of AFOLU estimates in the Assessment Reports relates to the use of
the bookkeeping model to estimate land use, land use change and forest (LULUCF)
emissions. As useful as this model is, its framework does not follow the IPCC AFOLU
guidelines (IPCC, 2006), particularly regarding the concept of managed land. Thus, forests
that are on managed land but are not suffering from direct human activities are considered
carbon neutral (Houghton *pers. comm.*). Partly because of that, the net emission estimates of
LULUCF from Houghton et al., (2012) used in the AR5 (4.03 $PgCO_2.yr^{-1,}$ 2010) contrast with
the LULUCF estimates produced by country reports submitted to the UNFCCC for the same
year, which are close to zero (Grassi and Dentener, 2015). The use of IPCC compliant models
for the IPCC Assessment Reports, or/and some documentation that warned about these
inconsistencies, would be useful in future assessments.




Emissions in the agricultural sector are mostly net, since sink effects in the soils are small and
frequently temporal (USEPA, 2013; Smith et al., 2014). Comparisons against global
agricultural emissions show that for the year 2000, global estimates more than doubled our
values (e.g. 5 and 5.5 $PgCO_2e.yr^{-1}$ *vs* ca. 2 $PgCO_2e.yr^{-1}$ in all datasets) (Tubiello et al., 2015)
(Table 2), suggesting larger contributions of agricultural emissions from non-tropical
countries. Unexplained methodological differences such as the inclusion or not of indirect
emissions and the lack of an exhaustive list of the variables included in the agricultural
emissions, difficult further comparisons.

### 3.2 Disaggregated gross emissions: contributions of the emission sources
While the gross aggregated estimates suggested a good level of agreement among datasets
(Figure 1), differences occur when comparing the emissions sources leading the AFOLU
budgets (Figure 2). The FOLU sector showed the largest differences, mainly due to the
estimates of forest degradation, and particularly fire (FAOSTAT and EDGAR showed the
lowest and highest values). The forest sector is the most uncertain term in the AFOLU
emissions due to both uncertainties in areas affected by land use changes and other
disturbances, and by uncertain forest carbon densities (Houghton et al., 2012; Grace et al.,
2014; Smith et al., 2014). Agricultural sources were more homogeneous (ca. 2 $PgCO_2e.yr^{-1}$
for all datasets) (Figure 1), with livestock and cropland soil emissions as the most and least
similar (Figure 2). The homogeneity in livestock emissions was expected since most datasets
use common statistics (FAO) to derive herd numbers per country.

### 3.2.1 Deforestation
Deforestation emissions were 2.9 (1.0-10.1), 3.7, and 2.5 and 4.2 $PgCO_2.yr^{-1}$ (Hotspots,





FAOSTAT, EDGAR, and Baccini, respectively), with Baccini and EDGAR showing the
highest and the lowest values. Their values represent, however, very different scenarios: gross
deforestation for the Hotspots and Baccini datasets, net deforestation for FAOSTAT, and
forest fire and post-burn decay for EDGAR (Table 3). The Hotspots (Harris et al., 2012) and
Baccini et al., (2012) datasets offer gross deforestation estimates that rely on Hansen et al.,
(2010)'s forest cover loss areas. However, they report different tropical emissions (0.81 and
1.14 PgC.yr$^{-1}$) because they use different carbon density maps: Harris *et al.*(2012) rely on
Saatchi *et al.*(2011) and Baccini rely on Baccini *et al.*(2010). EDGAR does not provide a
category for deforestation, and their Forest Fire and Decay category (5F) (Table 3, and Table
S1 in SI) is used as a proxy for deforestation (Tubiello et al., 2015). Such an approximation
leads to underestimations since not all carbon losses from deforestation are necessarily
associated with the use of fire (Tubiello et al., 2015). In spite of being net emissions, the
deforestation estimates for FAOSTAT were higher than the gross estimates of Hotspots and
Baccini. This is partly due to FAOSTAT's inclusion of fire emissions from humid tropical
forests (see section 3.2.3), which the other datasets did not. Baccini's larger estimates of gross
deforestation included more carbon pools than the other datasets (e.g. soil, CWD, litter).
Baccini *et al.* (2012) reported that their estimated gross and net emissions from tropical
deforestation were the same value (4.2 Pg $CO_2$ yr$^{-1}$). The difference with Houghton's net
emissions (5.4 Pg$CO_2$.yr$^{-1}$) (Figure 2) corresponds, then, to non-offset carbon emissions from
other land uses and activities included in the bookeeping model: degradation by logging and
shifting cultivation, decomposition and decay, and cultivated soils. Houghton's tropical net
emissions for 2000-2005 are high, but lower than Houghton's reported net estimates in the
80's (7 Pg$CO_2$.yr$^{-1}$) (Houghton, 1999).




*3.2.2 Forest degradation*
Forest degradation can be defined in many ways (Simula, 2009), but no single operational
definition has been agreed upon by the international community (Herold et al., 2011a). It
typically refers to a sustained human-induced loss of carbon stocks within forest land that
remains forest land. In this study, similarly to Federici et al., (2015), we consider degradation
any annual removal of carbon stocks that does not account for deforestation, without temporal
scale considerations (e.g. time needed for disturbance recovery, or time to guarantee a
sustained reduction of the biomass). We assessed two major degradation sources: wood
harvesting and fire. Soil degradation is poorly captured in many datasets, and mainly focuses
on fire in equatorial Asian peatland forests and drained peatlands (Hooijer et al., 2010).
Better understanding of the processes and emissions behind forest degradation, would be key
for climate mitigation efforts not only because forest degradation is a wide spread
phenomenon (e.g. affects much larger areas than deforestation (Herold et al., 2011b)) but also
because the lack of knowledge of net carbon effects frequently results in assumptions of
carbon neutrality of the affected standing forests, particularly for fire (Houghton et al., 2012;
Le Quéré et al., 2014), which is likely leading to an underestimation of forest and AFOLU
emissions.

Gross emissions from forest degradation were larger than deforestation for the Hotspots,
EDGAR and Baccini's datasets, with degradation-to-deforestation ratios of 108%, 120%, and
128%, respectively. FAOSTAT had degradation emissions of 60% of the deforestation, partly
due to its anomalously low fire contribution (see next section). Houghton et al., (2012)
pointed out that global FOLU net fluxes were led by deforestation with a smaller fraction
attributable to forest degradation, while the opposite was true for gross emissions (degradation
being 267% of deforestation emissions). This large ratio relates to their inclusion of shifting





cultivation under degradation. This is a definition issue, which would not fit the definition of
degradation chosen in this study, where a complete forest cover loss would represent
deforestation and not degradation.

*3.2.3 Fire*
Fire led the gross forest degradation emissions in the tropics in 2000-2005 (Figure 2): 2 (1.1-
2.7), 0.2, 3.4, 2.9 $PgCO_2e.yr^{-1}$ for the Hotspots, FAOSTAT, EDGAR, and Baccini datasets,
respectively) (Figure 2). Our estimates are conservative compared to Van der Werf et al.,
(2010)'s global emissions of 7.7 $PgCO_2e.yr^{-1}$ for 2002-2007, due to our removal of $CO_2$ from
deforestation fires (to avoid double counting with deforestation emissions), to the exclusion of
fires in grasslands and agricultural residues, and to our smaller study area. FAOSTAT and
EDGAR had the lowest and the highest fire values. FAOSTAT lowest values relate to
omissions that are currently in the process of being corrected (Rossi *pers. comm.*): 1. the
exclusion of $CO_2$ from fire in humid tropical forests and other forests (Table 3, Table S1),
which FAOSTAT relocated as net forest conversion emissions, partly explaining their larger
deforestation values, and 2. The use of default parameters for fuel in peats from the IPCC
2006 Guidelines instead of the new IPCC Wetland supplement which offer considerable
higher values (Rossi et al., 2016). Moreover, FAOSTAT uses GFED3.0-burned area (Giglio
et al., 2010) in their estimates while the other datasets use GFED3.0-emissions (Van der Werf
et al., 2010). EDGAR fire emissions were the largest most likely because they included decay.
Their dataset considers some undefined "forest fires" (5A) and "wetland/peatland fires and
decay" (5D) (Table 3; Table S1 in SI). Peatland decay probably explains EDGAR's larger
emissions in Asia, while we assume that EDGAR's highest fire emissions for CS America
might respond to deforestation fires which were not included in the Hotspots to avoid double
counting with deforestation, and relocated in FAOSTAT to deforestation emissions (Figure 3,



Table 3). Our Hotspots dataset showed higher gross fire emissions for Africa due to the inclusion of woodland fire, which EDGAR and FAOSTAT probably excluded. Baccini et al., (2012)'s fire emissions: 2.9 $PgCO_2e.yr^{-1}$ (2000-2010) derive from Houghton's bookkeeping but it is unclear how these emissions were estimated.

In spite of the importance of fire as a degradation source, this variable is frequently incompletely included, either through unaccounted gases (e.g. $CH_4$ and $N_2O$ are excluded in the carbon community but their omission represent 17-34% of the gross $CO_2$ fire emissions) (Valentini et al., 2014; Roman-Cuesta et al., under review), or to unaccounted components (e.g fires in tropical temperate forests such as conifers or dry forests such as woodlands, are frequently excluded) (Houghton et al., 2012). Unaccounted fire emissions also derive from methodological choices (e.g. only inter-annual fire anomalies being considered) (Le Quéré et al., 2014), from poor satellite observations such as understory fires in humid closed canopy forests) (Alencar et al., 2006; 2012, Morton et al., 2013), or satellite fire omissions in certain regions (e.g. high Andean fires) (Bradley and Millington, 2006; Oliveras et al., 2014). Other omissions relate to the current exclusion of non-Asian peatland fires (e.g American tropical montane cloud forest peatland fires) (Asbjornsen et al., 2005; Roman-Cuesta et al., 2011; Oliveras et al., 2013; Turetsky et al., 2015).

Fire suffers, moreover, from a series of assumptions that do not apply so easily to other types of degradation: 1. Assuming a non-human nature of the fires (deforestation fire *vs* wildfires), which in tropical areas contrasts with multiple citations referring to the 90% human causality of fires (Cochrane et al., 1999; Roman-Cuesta et al., 2003; Alencar et al., 2006; Van der Werf et al., 2010). 2.Assuming *force-majeure* conditions that lead to non-controllable fires due to extreme climate conditions, which frequently results in incomplete assessment and reporting of emissions. This assumption contrasts with research on how human activities have seriously




increased fire risk and spread in the tropics (Uhl and Kauffman, 1990; Laurance and
Williamson, 2001; Roman-Cuesta et al., 2003; Hooijer et al., 2010), and clearly expose how
most of the fires in the humid tropics would not occur in the absence of human influences
over the landscape (; Roman-Cuesta et al., 2003). 3. Assuming carbon neutrality and full
biomass recovery after fire in standing forests. This is a generous assumption that contrasts
with numerous studies on tropical forest die-back following fire events in non-fire adapted
humid tropical forests (Cochrane et al., 1999; Barlow et al., 2008; Roman-Cuesta et al., 2011;
Brando et al., 2012; Oliveras et al., 2013; Balch et al., 2015).  All these phenomena casts
doubts on the robustness of these assumptions and call for a much more comprehensive
inclusion of fire emissions into forest degradation budgets.

*3.2.4 Wood harvesting*
There is not a unique way to estimate wood harvesting emissions as exposed in the guidelines
for harvested wood products of the IPCC (IPCC 2006). Assumptions regarding the final use
of the wood products, decay times, substitution effects, international destination of the
products and time needed for forests to recover their lost wood, can fully change the emission
budgets. In out study, wood harvesting emissions were 1.2 (0.7-1.6), 2.0, 1.7 $PgCO_2.yr^{-1}$ for
the Hotspots, FAOSTAT and Baccini data, respectively (Tables 3, Table S1 in SI). Harvested
wood products derive from FAO's country reports (e.g. FAOSTAT forest products). All
datasets included fuel wood and industrial roundwood (Tables 3, Table S1). EDGAR
excluded fuelwood from the AFOLU budget and placed it instead into the energy budget
(EDGAR, 2012), which explains its absence in Figure 2. Wood harvesting emissions were
larger in FAOSTAT than in the Hotspot data (Figure 2) partly due to the inclusion of some
extra categories of fuels (e.g. charcoal and residues) that were not included in the Hotspot
database (Table 3, Table S1 in SI). Charcoal represents 26% of the total wood-harvesting



496 emissions in FAOSTAT. Differences on wood harvesting affected more Asia and CS America

497 (where our Hotspot data were half of FAOSTAT's), whilst Africa presented almost identical

498 values (Figure 3), reasons for these continental differences are unclear. Baccini's high

499 emissions on wood harvesting could partly relate to their inclusion of extra biomass due to

500 felling damages (e.g. 20-67% of the AGB is damaged, and 20% is left dead in BGB)

501 (Houghton, 1999).

502

503 *3.2.5 Livestock*

504 Livestock emissions were the most homogeneous among the emissions sources (Figure 2)

505 with estimates of 1.2 (0.8-1.5), 1.1, 1.2, 1.1 $PgCO_2e.yr^{-1}$ for the Hotspots, FAOSTAT,

506 EDGAR and EPA respectively, in range with the estimates in the AR5 (Fig 11.5 in Smith et

507 al., 2014). Values were similar in spite of deriving from different Tiers (e.g. Tier 3 for Herrero

508 et al., (2013), Tier 1 for FAOSTAT and EDGAR. EPA used Tier 3 but for incomplete data

509 series, otherwise Tier 1 was applied (USEPA, 2013)). All datasets included enteric

510 fermentation ($CH_4$) and manure management ($N_2O$, $CH_4$). All of them relied on FAO data for

511 livestock heads, although they used different years (e.g. 2000 for Herrero et al., (2013) data in

512 the Hotspots, and 2007-2010 for EDGAR). From a continental perspective, FAOSTAT and

513 EDGAR estimates were the closest while the Hotspots and EPA's were less similar. The

514 Hotspots showed higher emissions for Africa and Asia and lower for CS America, than the

515 other three datasets. Divergences likely relate to different Tiers. CS America and Asia showed

516 the highest values, with Africa following closely (Figure 3), similar to what is reported in the

517 AR5 (Smith et al., 2014). Globally, livestock is the largest source of $CH_4$ emissions, with

518 three-fourth of the emissions coming from developing countries, particularly Asia (USEPA,

519 2013, Tubiello et al., 2014). Three out of the top-5 emitting countries are in the tropics:



Pakistan, India and Brazil (USEPA, 2013) and while Asia hosts the largest livestock
emissions, the fastest growing trends in 2011 correspond to Africa (Tubiello et al., 2014).

*3.2.6 Cropland emissions*
The estimates of cropland emissions reached values of 0.18 (0.16-0.19), 0.56, 0.6 and 0.64
$PgCO_2e.yr^{-1}$ for the Hotspots, FAO, EDGAR and EPA datasets respectively, for $N_2O$ and $CO_2$
from changes in soil organic carbon content. Cropland soil emissions ($N_2O$ and soil organic
carbon stocks ($CO_2$)) heavily depend on land management practices (e.g. tillage, fertilization
and irrigation practices) and climate (Crowther et al., 2015). We chose exactly the same land
practices in all datasets to allow comparisons (Table 3,S1 in SI). For this reason, we excluded
$N_2O$ emissions from grassland soils, drainage of organic soils, and restoration of degraded
lands (Table 3). This restrictions resulted in lower emissions than those estimated for cropland
soils in the AR5 (Fig. 11.5 in Smith et al., 2014). The Hotspots and EPA showed the lowest
and the highest estimates (Figures 2,3). With the exception of the Hotspots, the other datasets
agreed well at the tropical scale, with FAOSTAT and EDGAR being almost identical, also at
continental scales. EPA disagreed more than the other datasets at the continental scales, with
underestimations for Asia, probably related to the parameterization of their emission model.
All three datasets used FAO's activity data, and for EDGAR and FAOSTAT the same
emission factors must have been used. The Hotspot showed anomalously low emissions partly
because it only included six major crop types (maize, soya, sorghum, wheat, barley, millet)
for which the emission model (DAYCENT) counted on reliable parametrization (*Ogle pers.*
*comm*). Emissions from other important crops in the tropics (e.g. sugar cane, tobacco, tea, etc)
were excluded, as well as emissions from croplands in organic soils, due to model constraints.

*3.2.7 Peatland drainage for agriculture*



The disaggregation of cropland soil emissions from drained peatands shows large omissions
for drained peatlands in the Hotspots database. Emissions were one order of magnitude lower
(28 $TgCO_2e.yr^{-1}$) than FAOSTAT (ca. 500 $TgCO_2e.yr^{-1}$) and than the peatland drainage
emissions reported in Asia alone by Hooijer et al. (2010) (355-855 $TgCO_2e.yr^{-1}$) Our lower
values relate to much smaller agricultural areas with histosols (0.4 mill ha) than those reported
by FAOSTAT for the same countries (7mill ha). Differences relate to the subset of the final
areas to only those that respond to the six types of crops selected by Ogle et al. (2013) (maize,
wheat, sorghum, soya beans, millet and  barley), to the unmatching spatial scales of the
overlapping layers (1km for histosols and 50km for croplands) which result in
underestimations of the final area, and to the use of an Emission Factor of 20 $MgC.ha^{-1}$ for the
Hotspots data, while FAOSTAT used 14.64 $MgC.ha^{-1}$.

*3.2.8 Paddy rice*

When paddy fields are flooded, decomposition of organic material gradually depletes the
oxygen present in the soil and floodwater, causing anaerobic conditions in the soil that favour
methanogenic bacteria that produce $CH_4$. Some of this $CH_4$ is dissolved in the floodwater, but
the remainder is released to the atmosphere, primarily through the rice plants themselves. Net
emission estimates for paddy rice were 0.55 (0.4-0.833), 0.33, 0.37, 0.30 $PgCO_2e.yr^{-1}$ for the
Hotspots, FAOSTAT, EDGAR and EPA datasets, respectively. The Hotspots showed the
highest emissions (Figure 2), but only in Asia (Figures 3). Part of the reason behind these
differences refers to the final gases estimated in Li et al., (2013)'s which included $CH_4$, $N_2O$
and SOC ($CO_2$) (Table 3, S1), while the others only focused on $CH_4$. In Li et al., (2013)'s
estimates, $N_2O$ were 48% of the $CH_4$ emissions, explaining the doubled emissions in our
database. SOC was a sink, with -0.076 $PgCO_2.yr^{-1}$.



Based on the above, Table 4 offers the least reliable emission sources of each dataset.

*3.3 Differences in the relative contribution of greenhouse gases (CO$_2$, CH$_4$, N$_2$O)*
GHG emissions (CO$_2$, CH$_4$, N$_2$O) showed good agreement at the sectoral level (FOLU and
agriculture) (Figure 5), that disappeared at the disaggregated level (Figure 6). CO$_2$ showed the
largest disagreements among datasets and gases, led by forests emissions and particularly fire.
SOC accumulation was reported in the Hotspots data (Li et al., 2013) but it is uncertain if it is
included in the other datasets.

Non-CO$_2$ emissions were much more homogeneous, with differences among datasets that
were approximately 5 times lower than CO$_2$ variability (e.g. 0.3 vs 1.5) (Figure 6a). Livestock
led CH$_4$ emissions and showed the largest differences among datasets, with the Hotspot data
(Herrero et al., (2013) having the lowest CH$_4$ emissions, which were compensated with larger
N$_2$O than the other datasets (Figure 6b,c).
At a global level, wetlands dominates natural CH$_4$ emissions, while agriculture and fossil
fuels represent 2/3 of all human emissions, with smaller contributions coming from biomass
burning, the oceans, and termites (Montzka et al., 2011). Fire non-CO$_2$ emissions were quite
similar among datasets, confirming that FAOSTAT omissions were CO$_2$ related. Thus, as
exposed in FAOSTAT's metadata,  only N$_2$O and CH$_4$ are considered in forest fires,
excluding CO$_2$ from aboveground biomass.  As expected, N$_2$O emissions in crops showed
large differences, with our Hotspots having the lowest values (3 times lower). Rice N$_2$O
emissions were omitted in all datasets except the Hotspots (Li et al., 2013), which also
included SOC.



The importance of multigas assessments relates to their role in radiative forcing (RF)
understood as a measure of the warming strength of different human and natural agents (gases
and not gases) in causing global warming ($W.m^{-2}$). $CO_2$ is the most abundant 379 ppm in
2005 (400ppm in 2015), leading to an RF of $1.66\pm0.17$ $Wm^{-2}$. Fossil fuels and cement
production have contributed about three-quarters of that RF, with the remainder caused by
land use changes (AR4). The growth rate of $CO_2$ in the atmosphere in 1995-2005 (1.9 ppm.
$yr^{-1}$) increased the $CO_2$ RF by 20%, being the largest change observed or inferred for any
decade in the last 200 years (AR4). Non-$CO_2$ GHG are less abundant in the atmosphere
(1,774 ppb and 319 ppb for $CH_4$ and $N_2O$ in 2005 respectively) but have larger warming
potentials (x 28 for $CH_4$) and (x 265 for $N_2O$) ($0.48\pm0.05$ and $0.16\pm0.02$ $Wm^{-2}$ in 2005,
respectively) (AR4) and shorter lifetimes than $CO_2$ (~9 and ~120 years, respectively) offering
an additional opportunity to lessen future climate change (Montzka et al., 2011). Growth rates
in the atmosphere differ among gases with $CO_2$ and $N_2O$ showing quasi linear increases while
$CH_4$ shows peculiar patterns that are not fully resolved (Montzka et al., 2011). The sensitivity
of $CH_4$ emissions from wetlands to warmer and wetter climates suggests a positive feedback
between emissions and climate change that is visible in ice-core records (Montzka et al.,
2011). In the case of $N_2O$, and contrarily to the large contribution of non-human $CH_4$
emissions, anthropogenic emissions currently account for most of them (40%) primarily from
agricultural activities.

*3.4 Country level emissions*
Figures 7 and 8 show country level agreement for the AFOLU, forests, cropland and livestock
emission sectors, for the  FAOSTAT, EDGAR and Hotspot databases. The use of percentiles
forced each figure to have a similar number of countries per category of agreement (high,
moderate, low and very low), in detriment of sectorial comparisons. Thus, if we contrasted



forests to livestock emissions, the later would have had most countries on the level of high
agreement. However, we thought it useful to search for within emissions differences, to
improve the estimates in each emission sector. No country had high agreement for all the
emission sectors, with Brazil, India and Cambodia showing the best results (high agreement in
3 out of 4 sectors). CS America (Mexico, Guatemala, Bolivia, Venezuela, Paraguay,
Argentina, Uruguay) and Asia (Myanmar, Viet Nam, Thailand, Indonesia, Malaysia) showed
the second best agreements (3 out 4 sectors with high or moderate agreement). No country
showed very low agreement for all the emission sectors, but African countries (Angola,
Botswana, Somalia, Nigeria, Ghana, Cote d'Ivoire), CS American (Chile, French Guiana,
Suriname) and Asian (Papua New Guinea, Sri Lanka and Nepal) showed the largest
disagreements (3 out of 4 sectors with low or very low agreement). From a sectorial
perspective, emissions showed good agreement where they were expected to peak (e.g. forest
agreement was high in tropical countries, livestock in Asia, crops in CS America and parts of
Asia) (Figures 7,8). From a continental perspective, Africa showed more countries with high
levels of disagreement, suggesting the need for further data research.

***3.5 Some reflections on the datasets***
*3.5.1 Original goals*
Different datasets were developed for different purposes that have influenced the methods and
approaches chosen to estimate their land use GHGs. Thus, EDGAR was created with an air
pollution focus making its land emissions weaker. Contrastingly, FAOSTAT carries FAO's
focus on land,  particularly agriculture, with forest data coming later, through the FRA
assessments. The 'Hotspot' database was created to identify the areas with the largest land use
emissions in the tropics (emissions hotspots), while Houghton's accent is on historical
LULUCF emission trends (since 1850). EPA concentrates on industrial, energy, and



agricultural emissions -forests are excluded- with an interest on human health and mitigation.
Moreover,  several datasets rely on FAOSTAT's long-term agricultural data, which probably
explains why the agricultural estimates are more homogeneous (crops, rice, and livestock).
FAOSTAT's forest emissions use FRA data, which get updated every 5 years. Different FRA
versions strongly influence forest emission  and must be considered when comparing
estimates (e.g. differences up to 22% between the forest sink estimates using FRA2015 and
FRA2010 have been reported by Federici et al., 2015). Similarly, different versions of
Houghton's bookkeeping TRENDS data, as well as researchers' self-tuned versions of his
model, result in emission differences that are difficult to track.

*3.5.2 IPCC guidelines and guidance:* Under the UNFCCC, countries are requested to use the
latest IPCC AFOLU guidelines to estimate their GHG emissions (e.g. IPCC 2006 and 2003
for developed and developing countries, respectively). The use of different guidelines, Tiers,
and approaches influences the final emission estimates. Compliance with IPCC has two main
consequences: 1. the total area selected to report emissions, and 2. the choice of *land use* over
*land cover*.  In the first case, under IPCC guidance, the total area selected to report emissions
would include all the land under human influence (the *managed land* concept, which includes
areas under active and non-active management). Houghton's bookkeeping model (and the
carbon modelling community in general) do not comply well with the *managed land* concept,
resulting in different net emissions from forest land uses and land use changes (LULUCF)
than IPCC compliant country emissions (Grassi and Dentener, 2015; Federici et al., 2016). In
the second case, the selection of *land uses* instead of *land covers* has partly been behind the
recent controversy between FAO and the Global Forest Watch's reported estimates on
deforestation trends (REF). Estimates of deforestation that rely on *land cover* are higher than
those using *land use*, since forest losses under forest land uses -that remain forest land use-



are not considered deforestation (e.g. logged areas will regrow). In our analysis, FAO and

Houghton relies on *land use* for deforestation, while the 'Hotspots' and EDGAR rely on *land*

*cover*. FAOSTAT and the 'Hotspots' rely on the 2006 IPCC Guidelines for National

Greenhouse Gas Inventories (IPCC, 2006). FAOSTAT uses Tier 1 and standard emission

factors, while the 'Hotspots' use a combination of Tiers (Tier 3 for all emissions except wood

harvesting and cropland emissions over histosols that rely on Tier 1). EDGAR reports the use

of 2006 IPCC Guidelines for the selection of the emission factors but some of their

methodological approaches are not always consistent with IPCC guidelines (e.g. deforestation

expressed as the decay of burned forests, wood-harvesting is part of the energy sector,

agricultural energy balances are included in the AFOLU budget). EPA methods are reported

to be consistent with IPCC guidelines and guidance, with Tier 1 methodologies used to fill in

missing or unavailable data (USEPA, 2013).

**4. CONCLUSIONS**

The Paris Agreement (COP21) counts on the Nationally Determined Contributions (NDCs) as

the core of its negotiations to fight climate change. As March 2016, 188 countries had

submitted their NDCs under the UNFCC (FAO, 2016) with agriculture (crops, livestock,

fishery and aquaculture) and forests as prominent features in meeting the countries' mitigation

and adaptation goals (86% percent of the countries include AFOLU measures in their NDCs,

placing it second after the energy sector) (FAO, 2016). However, there exists large variability

in the way countries present their mitigation goals, and quantified sector-specific targets are

rare (FAO, 2016). Variability relates not only to the lack of a standardized way to report

mitigation commitments under the NDCs, but also to uncertainties and gaps in the AFOLU

data. The Paris Agreement relies on a 5-year cycle stock-taking process to enhance mitigation

ambition, and to keep close to the 2°C target. To be effective and efficient, stock-taking needs



robust, transparent and certain numbers (at least with known uncertainties). This is true both
for national emission reports and NDCs, but also for the global datasets that can be used to
review the feasibility of countries' mitigation claims, and the real space for further mitigation
commitments. We have here compared the gross AFOLU emissions of six datasets to search
for disagreements, gaps, and uncertainties, focusing on the tropical region. Conclusions
depend on the spatial scale. At the tropical scale:
- Data aggregation offers much closer emission estimates than disaggregated data (e.g.
country level, continental level, gas level, emission source level).
- Forest emissions are the most uncertain of the AFOLU sector, with deforestation
having the highest uncertainties.
- Agricultural emissions, particularly livestock, are the most homogeneous of the
AFOLU emissions.
- Forest degradation, both fire and wood harvesting, show the largest variabilities
among databases.
- $CO_2$ is the gas with longer-term influence in climate change trends, but remains the
most uncertain of the AFOLU gases.
- In absolute values, GHG disaggregation shows the largest differences for $CO_2$ in fire
emissions.
- $N_2O$ variability affected all the emission sources, making it the most dissimilar of the
non-$CO_2$ gases.
- Emissions from histosols/peatlands remain incomplete or fully omitted in most
datasets.
At continental level:
- The level of disagreement of the emission sources at continental scale makes it
difficult to track the most possible drivers behind the emissions.



At country level:
• Countries with higher agreement among databases were present in all continents, with
Africa showing the highest levels of country disagreement.

***4.1 Next steps***
*4.1.1 Enhancing dialogue between the carbon and the AFOLU research communities*
Research ran by the carbon community is pivotal for AFOLU assessments and while these
two research communities overlap, they do not focus on exactly the same topics. The carbon
community works with $CO_2$ emissions-only, fully excluding non-$CO_2$ gases, particularly $N_2O$.
It moreover rather focuses on forests and associated land use changes, excluding emissions
from agriculture. The AFOLU community has, contrarily, a multi-gas approach ($CO_2$, $CH_4$,
$N_2O$) and includes emissions from both forests and agriculture. For these reasons, estimates
of the carbon community cannot be considered as AFOLU estimates, and certain confusion
appears in the IPCC's AR5 with an incorrect AFOLU labelling (Table 11.1, Fig S2 in SI).
There is great space for these two communities to cooperate but further dialogue is needed to
promote closer and more coordinated action. Future steps might include the adoption of the
*managed land* concept by the carbon community; and ways to include legacy emissions by
the AFOLU community.

*4.2.2 Improving data quality*
The quality of the reported AFOLU emissions can be assessed through the UNFCCC
principles: completeness, comparability, consistency, accuracy and transparency, which can
help navigate the improvements of national monitoring systems. From these principles, the
reviewed datasets performed well in *consistency* (they applied similar methods and
assumptions over time, with the exception of 'Hotspots' that did not include temporal data).





*Transparency* was excellent for FAOSTAT with well elaborated and publicly available
metadata linked to their offered data, while EDGAR performed poorly due to insufficient
metadata. Improving transparency is an urgent call for future action. *Accuracy and*
*uncertainty* are also urgent calls. Thus, in spite of their importance to fully understand the
emission trends and dynamics, only Houghton and the 'Hotspots' provided uncertainties.
FAO offered uncertainties as a percent value for each emission source. *Completeness and*
*omissions* are also urgent tasks because all datasets are incomplete (Table 1) (e.g. missing
pools, missing gases) and omissions affect all datasets. Complete emission reporting should
consider the importance of:
• Forest soil $CO_2$ and $N_2O$ emissions (Werner et al., 2007) (e.g. $N_2O$ tropical forest soil
emissions of 0.7 $PgCO_2e.yr^{-1}$).
• Emissions from $CH_4$ and $N_2O$ from drained peatland soils, and from wetlands over
managed land (e.g. conservation).
• All forest fire types (e.g. temperate conifers and woodlands; understory fires over
humid closed canopy forests (Alencar et al., 2006; Morton et al., 2013) (e.g. 85,500
$km^2$, 1999-2010 in southern Brazilian Amazon); fire emissions over peatland soils and
peatland forests out of Asia (Román-Cuesta et al., 2011; Oliveras et al., 2014) (e.g. 4-8
$TgCO_2e$, 1982-1999, for the tropical high Andes from Venezuela to Bolivia)
• $CO_2$ emissions from other components of wood harvesting other than fuel and
industrial roundwood (e.g. charcoal, residues).
• $CO_2$ emissions from tree biomass loss due to fragmentation (Numata et al., 2010; Pütz
et al., 2014) (e.g 0.2 $Pg\ C\ y^{-1}$)
• $CO_2$ due to decomposition and decay of forests under extreme events: hurricanes
(Read and Lawrence, 2003; Negron-Juarez et al., 2010) (e.g the 2005 convective
storm, the Amazon basin suffered from an estimated tree mortality of 542±121 million





trees); intense droughts (Phillips et al., 2009, 2010; Brienen et al., 2015) (e.g. the 2005

Amazonian drought resulted in 1.2-1.6 PgC emissions and the atmosphere has yet to

see 13.9 $PgCO_2$ (3.8 PgC) of the Amazon necromass carbon produced since 1983);

Further suggestions on improving data gaps and knowledge for the AFOLU sector have been

reported by Smith *et al.* (2014); Houghton *et al.* (2012); USEPA (2013) and Sist *et al.* (2015),

with a focus on soil data and crop production systems, as well as an improved understanding

of the mitigation potentials, costs and consequences of land use mitigation options.

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

RMRC, MR, MH designed the study. SO, BP provided data and ran quality controls of the
data. RMRC, MR, MH, KBB, TR, LV, CM, SR, RH, SO, BP discussed the results and
contributed to writing.

**7. ACKNOWLEDGEMENTS**
This research was generously funded by the Standard Assessment of Mitigation Potential and





Livelihoods in Smallholder Systems (SAMPLES) Project as part of the CGIAR Research
Program Climate Change, Agriculture, and Food Security (CCAFS). Funding also came from
two European Union FP7 projects: GEOCarbon (283080) and Independent Monitoring of
GHG Emissions-N° CLIMA.A.2/ETU/2014/0008. Partial funds came through CIFOR from
the governments of Australia (Grant Agreement # 46167) and Norway (Grant Agreement
#QZA-10/0468). In the memory of Changsheng Li.



**Figures**

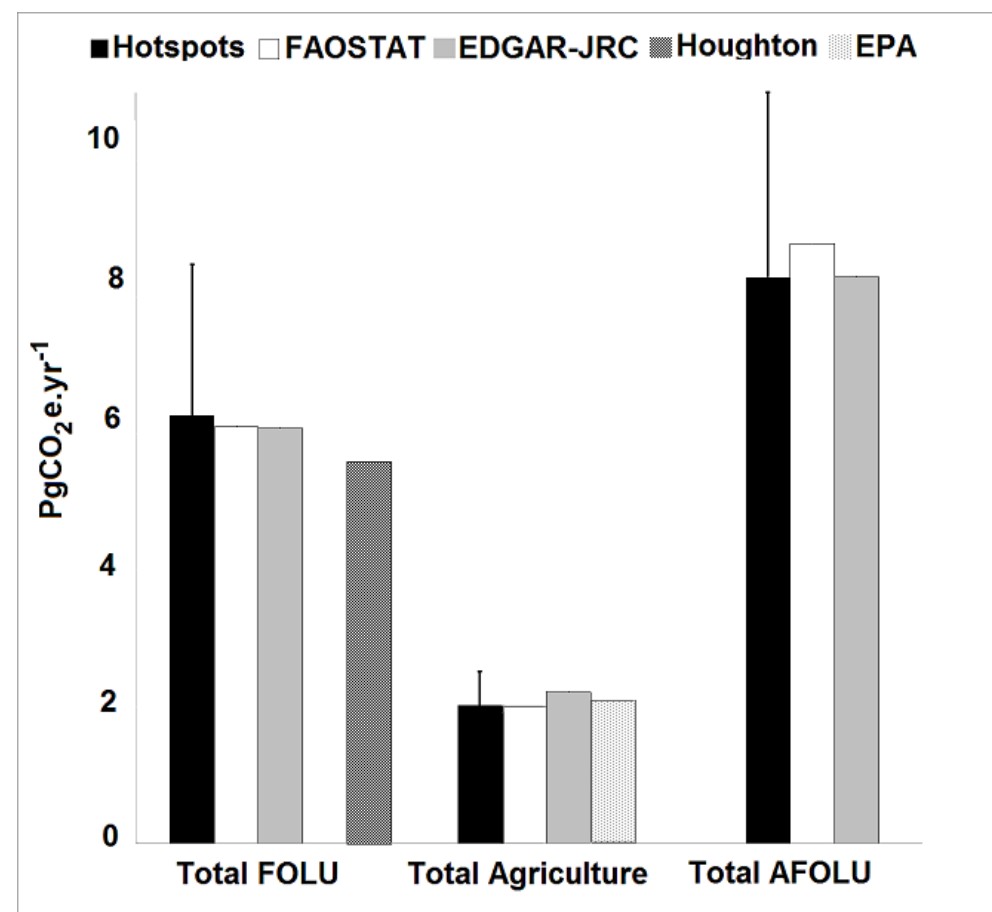



**Figure 1:** AFOLU tropical emissions estimates ($PgCO_2e.yr^{-1}$) for the period 2000-2005, for
five datasets (EDGAR, FAOSTAT, Hostpots, Houghton, EPA), disaggregated into FOLU
(Forestry and Other Land Use) and Agricultural emissions.




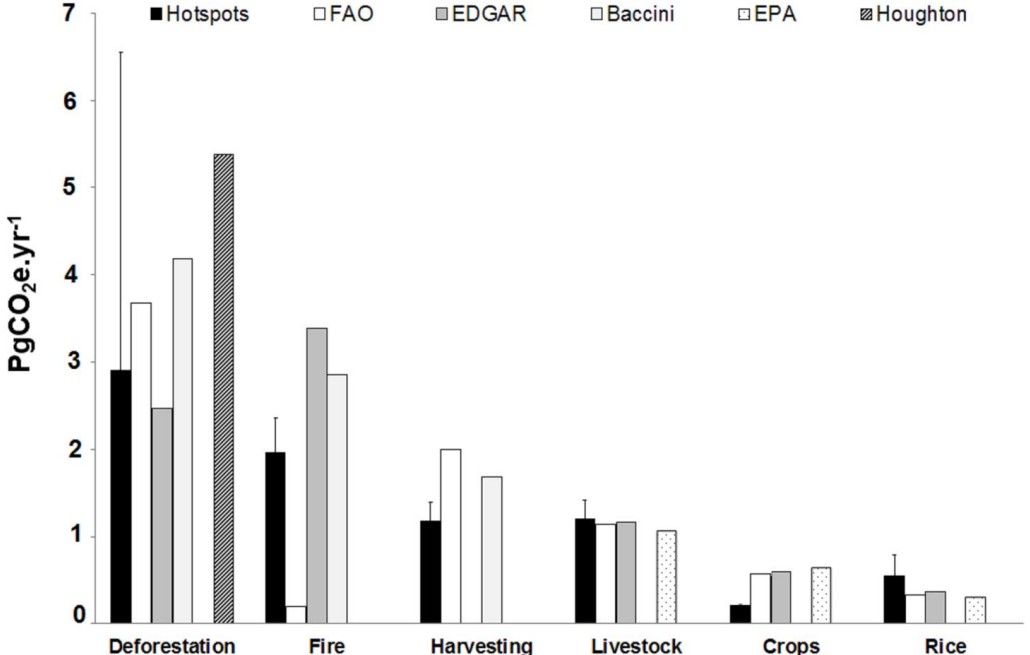


**Figure 2:** Tropical gross annual emissions (2000-2005) comparisons, for the leading emission

sources in the AFOLU sector, for the Hotspots, FAOSTAT, EDGAR, Baccini, EPA and

Houghton datasets, in this order.  Houghton's data are net land use emissions rather than

deforestation and are offered for visual comparisons against Baccini's gross deforestation

estimate.



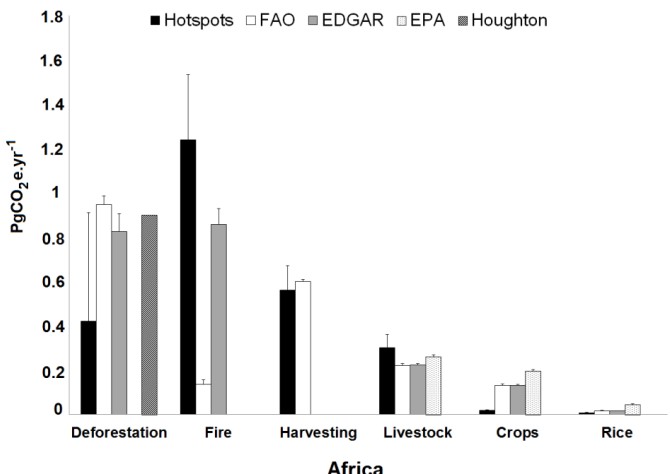

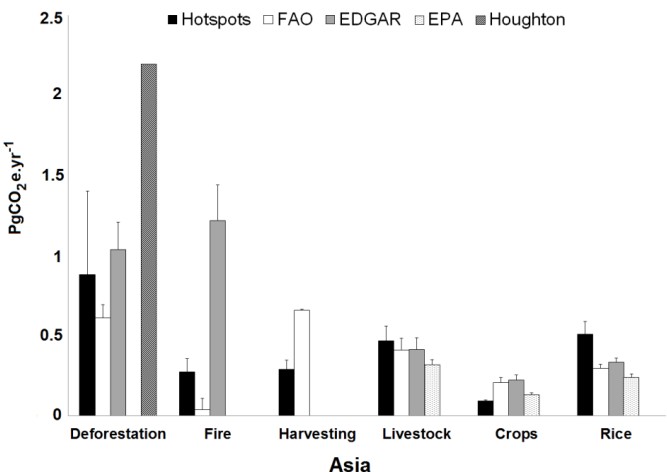

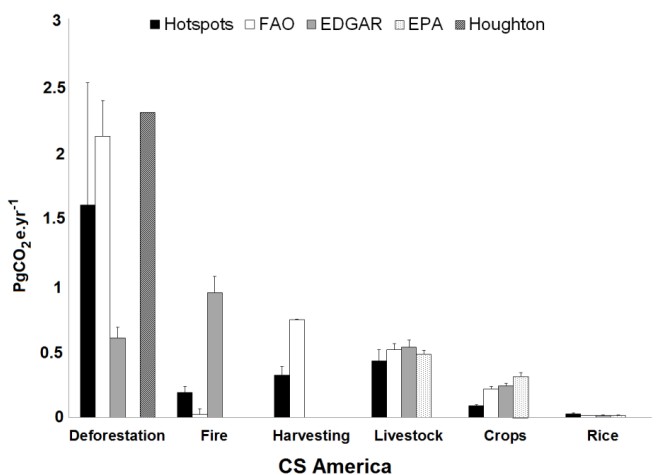





**Figure 3**: Continental disaggregated emissions for the individual emission sources in
$PgCO_2e.yr^{-1}$. Bars indicate uncertainty estimates ($1\sigma$ from mean). No uncertainty estimates
are available for the other datasets.





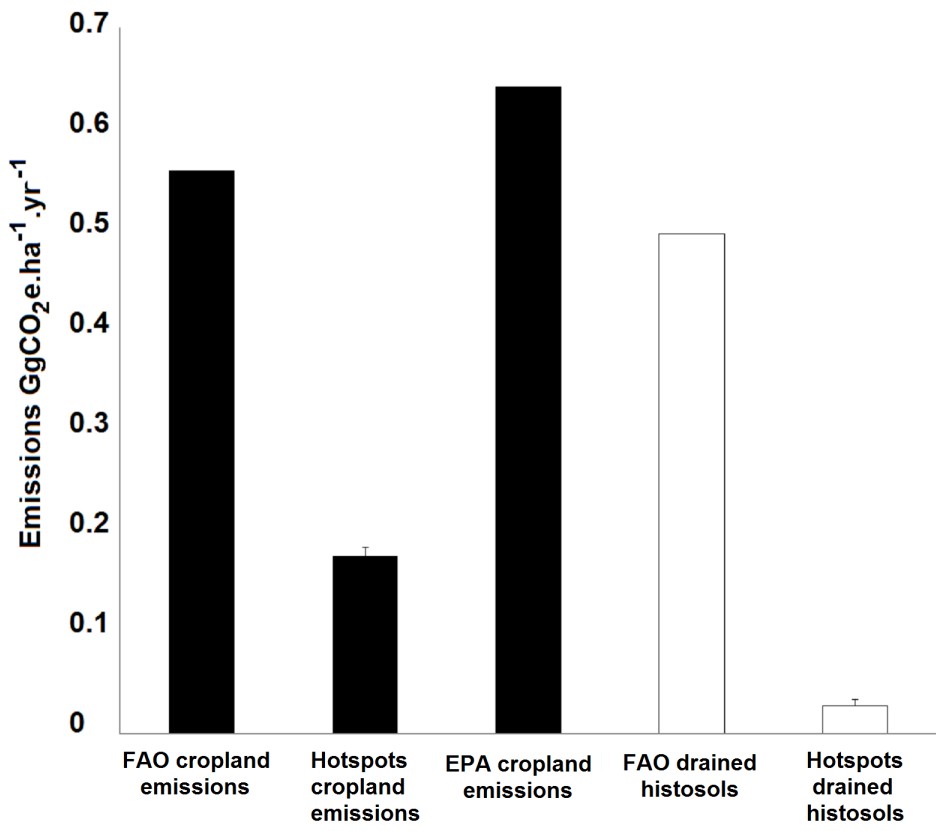


**Figure 4**: Disaggregation of cropland soil emissions from drained peatands for the datasets where data were available in a disaggregated manner (FAOSTAT and Hotspots). Organic soils were excluded in EPA's cropland emissions.






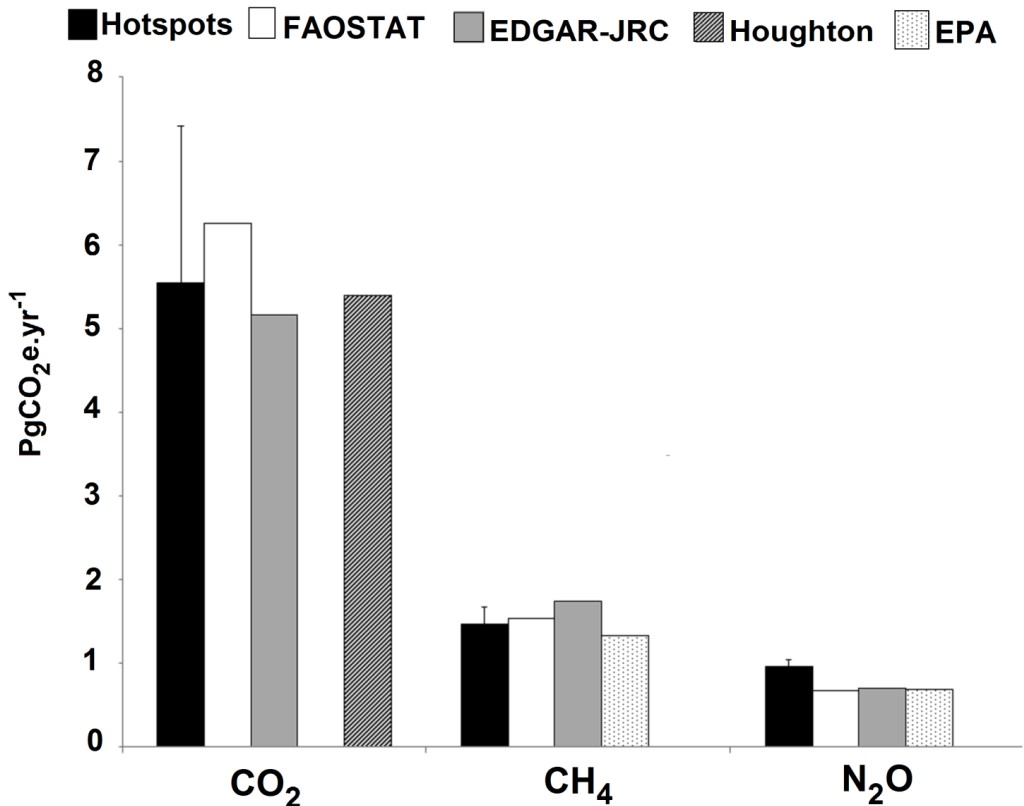


**Figure 5:** Contribution of the different AFOLU GHGs (CO2, CH4 and N2O) for the different

datasets. Bars indicate uncertainty estimates (1σ from mean). No uncertainty estimates are

available for the other datasets.



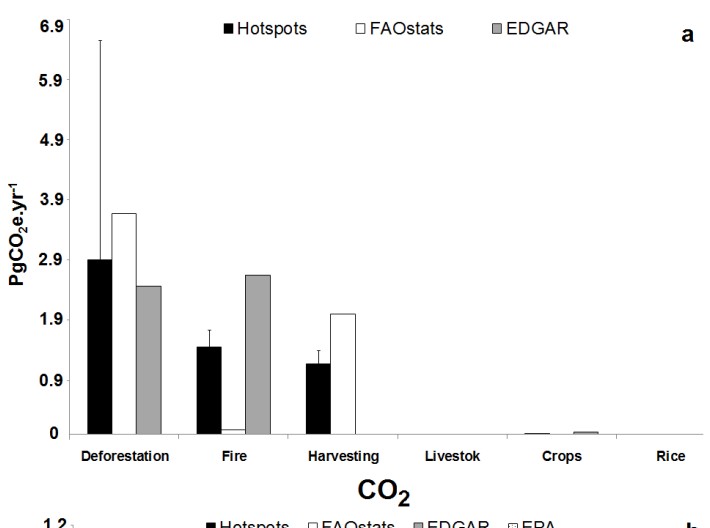

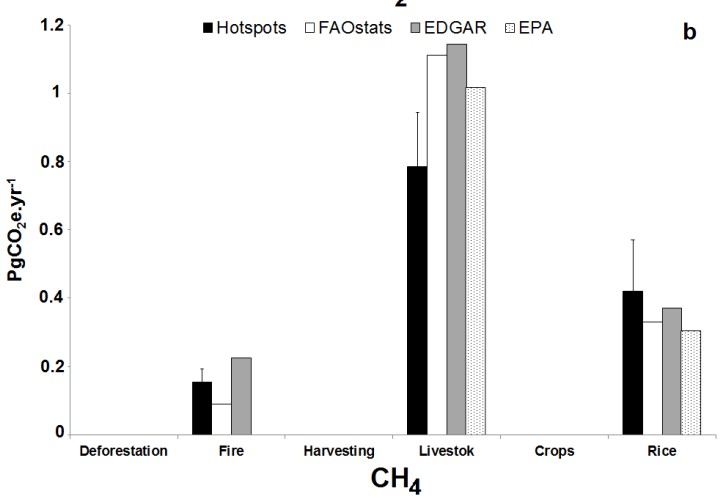

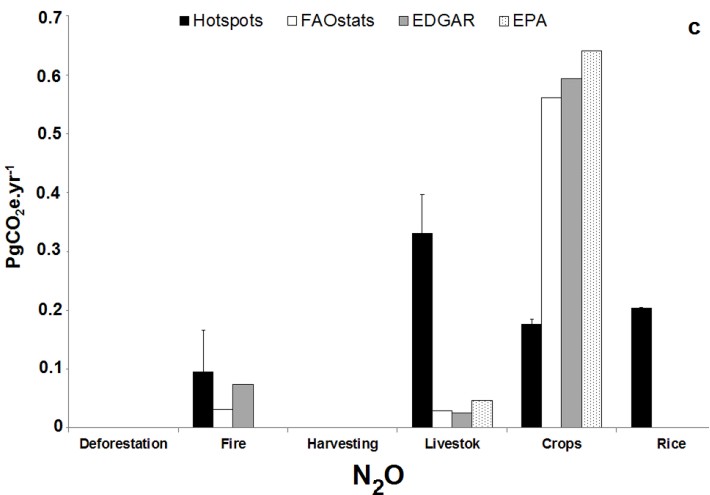





**Figure 6**: GHG emission contribution ($CO_2$, $CH_4$ and $N_2O$) of the leading AFOLU emission
sources. Bars indicate uncertainty estimates ($1\sigma$ from mean). No uncertainty estimates are

available for the other datasets.

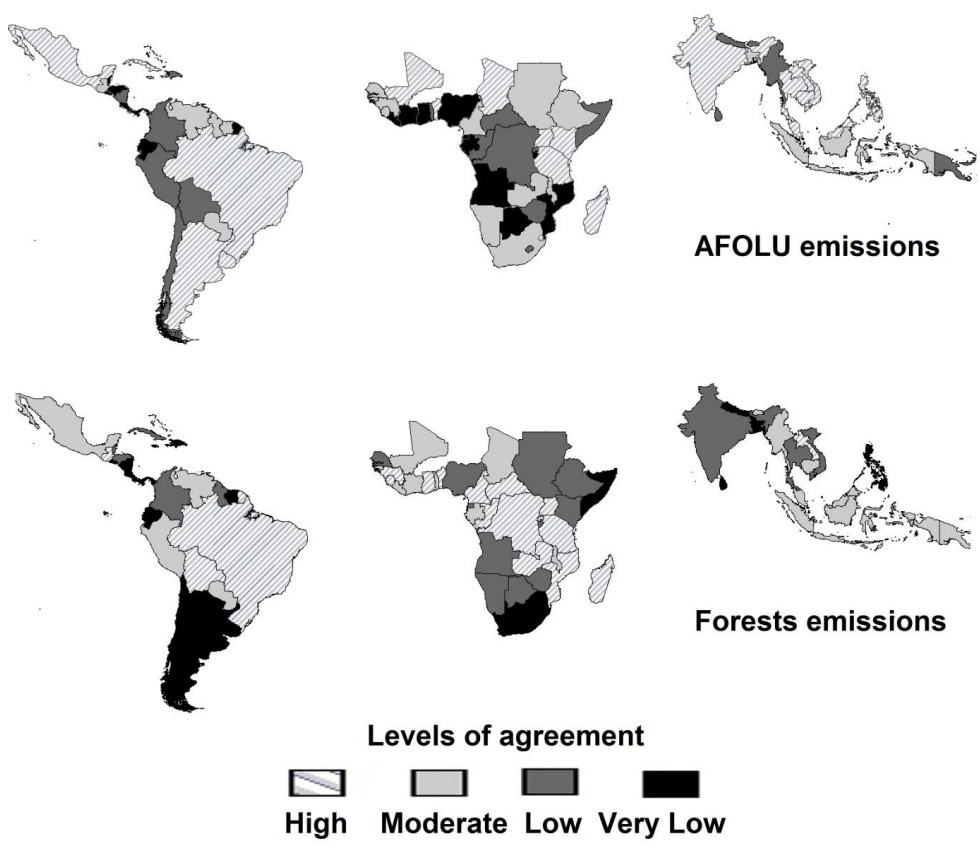


**Figures 7**: Country level agreement for AFOLU and forest emissions for the FAOSTAT,
EDGAR and 'Hotspots' databases. The categories of agreement are percentiles of the
coefficient of variation of the emission data (e.g. high agreement $\geq 75^{th}$ percentile, Moderate:
$50^{th}$ -$75^{th}$ percentiles, Low: $50^{th}$-$25^{th}$ percentiles, Very Low$\leq 25^{th}$ percentile)



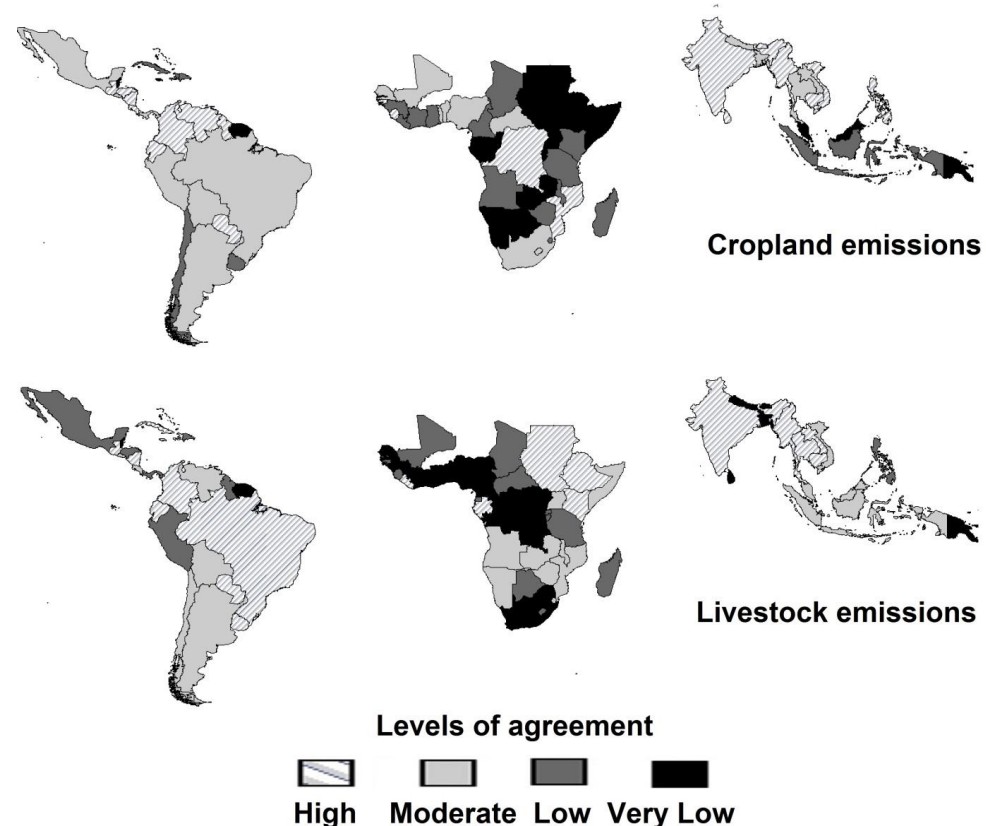


**Figures 8**: Country level agreement for croplands (cropland soils including histosols and rice)

and livestock emissions, for the FAOSTAT, EDGAR and 'Hotspots' databases. The

categories of agreement are percentiles of the coefficient of variation of the emission data

(e.g. high agreement $\geq 75^{th}$ percentile, Moderate: $50^{th}$ -$75^{th}$ percentiles, Low: $50^{th}$-$25^{th}$

percentiles, Very Low$\leq 25^{th}$ percentile)



**Tables**

| | Hotspots | FAOSTAT | EDGAR | Houghton | Baccini | EPA | AR5 |
|---|---|---|---|---|---|---|---|
| Gross/Net emissions | Gross | Gross | Gross | Net | Gross | Gross | Net |
| Uncertainty[a] | ✓ | No | No | No | No | No | ✓ |
| Transparency | High | High | Low[b] | Low | Low[d] | Intermediate | Low |
| IPCC compliant | ✓ | ✓ | ✓ | Not fully[c] | Not fully[d] | ✓ | Not fully[e] |
| Forest carbon Pools | AGB + BGB | AGB + BGB | AGB | AGB+BGB+Soil +CWD+Litter | AGB+BGB+Soil +CWD+Litter | Soil | AGB+BGB+Soil +CWD+Litter |
| Gases | $CO_2$,$CH_4$, $N_2O$ | $CO_2$,$CH_4$, $N_2O$ | $CO_2$,$CH_4$, $N_2O$ | $CO_2$ | $CO_2$ | $CO_2$,$CH_4$, $N_2O$ | $CO_2$ for forests. $CO_2$,$CH_4$, $N_2O$ for agriculture and peatlands. |
| Tier 1 | ✓ | ✓ | ✓ | ✓ | | ✓ | - |
| Tier 2, 3 | ✓ | ✓ | | | ✓ | ✓ | - |
| Spatial Disaggregation[f] | Pixel (0.5°) | Country | Country[g] | Region | Region | Country | Region |
| Peatlands | ✓ | ✓ | ✓ | No | No | No | ✓ |

**Table 1:** differences and similarities of the assessed AFOLU datasets.

a Uncertainty at the level of disaggregation at which data are available to download.

b Low means there is no metadata available, or metadata does not properly document the processes followed to estimate the emissions.

c EDGAR data on deforestation emissions does not follow IPCC guidelines.

d The bookkeeping approach does not follow the concept of managed land, and does not include the sink of forests remaining forests in managed land other than logged forests and those regrowing after shifting cultivation.





e Based on Houghton et al., (2012).
f Available disaggregated data.
g We selected data at the country scale to favour comparability with other datasets (e.g. FAOSTAT) even though data are available at pixel level
(0.1°).



| (b) | Net Global PgCO$_2$e.yr$^{-1}$ | | | | | | |
| --- | --- | --- | --- | --- | --- | --- | --- |
| | **2000** | | | **2010** | | | **2000/09** |
| | **FAOSTAT** | **EDGAR** | **Houghton** | **FAOSTAT** | **EDGAR** | **Houghton** | **AR5*** |
| **Agriculture** | 5 | 5.5 | - | 5.2 | 5.8 | - | 5 |
| **FOLU** | 4.9 | 6.5 | 4.9 | 4.9 | 5.5 | 4.2 | 5 |
| **AFOLU** | 9.9 | 12 | - | 10.1 | 11.3 | - | 10 |

**Table 2:** Summary of (a) tropical gross emissions estimates for agriculture, FOLU and AFOLU for all the datasets (Hotspots, FAOSTAT, EDGAR, EPA, Houghton) (2000-2005) and published data (Baccini et al., 2012, AR5 (Smith et al., 2014)) (2000-2007), and of (b) net global estimates as reported by Tubiello et al., (2015). Houghton and EPA offer FOLU and agricultural data only, respectively, and therefore estimates for AFOLU are not complete. *Data exposed in Figure 11.2 in Chapter 11 Smith et al. (2014).

*net FOLU flux estimate.

** Baccini et al., (2012) reported gross estimates for the FOLU components.

*** Baccini et al., (2012) estimates selected for the AR5 FOLU values in Figure 11.8, Chapter 11, WG-III.



| | Deforestation | Wood Harvesting | Fire | Enteric Fermentation | Manure management | Agricultural soils | Cropland over histosols | Rice | Others |
|---|---|---|---|---|---|---|---|---|---|
| $CO_2$ | 1[1],2[2],5[3],6[1] | 1[4],2[5],5[4],6[4] | 1[6],2[7], 3[8],5[9],6[9] | | | | 1[10],2[11] | | 3[12] |
| $CH_4$ | | | 1[13],2[14],3[15] | 1,2,3,4 | 1,2,3,4 | | | 1,2,3,4 | |
| $N_2O$ | | | 1[13],2[14],3[15] | | 1,2,3,4 | 1[16,17],2[16,18],3[16,17,19], 4[16,19] | 1,2 | 1 | |
| dSOC | | | | | | 1 | | 1 | |

**Table 3:** Contribution of different datasets to the different emission sources, disaggregated by GHG gases. 1: Hotspots, 2: FAOSTAT, 3: EDGAR, 4: EPA (only non-$CO_2$ agriculture emissions including livestock), 5: Houghton (only $CO_2$ FOLU emissions. No disaggregated data offered), 6: Baccini et al., 2012 (only $CO_2$ FOLU emissions, based on Houghton bookkeeping model). FAOSTAT are estimated through Tier 1 approaches.

[1] Gross deforestation.

[2] Net deforestation

[3] Houghton net $CO_2$-only estimates are not deforestation emissions, but land use and land use change fluxes including deforestation, forest degradation, and cropland, abandoned land, and agricultural soil organic carbon (SOC).

[4] Nationally reported fuel wood and industrial roundwood.

[5] Nationally reported fuel wood, charcoal, fuel residues and industrial roundwood.





[6] Long-cycle $CO_2$ emissions only (e.g savannas and agricultural $CO_2$ emissions are excluded). $CO_2$ emissions from peat, forests and woodland
fires (as defined by Van der Werf et al., 2010).
[7] $CO_2$ from the combustion of organic soils.
[8] $CO_2$ Forest fires + wetland/peatland fires and decay (5A, and 5D classes).
[9] Humid forest deforestation fires, and peatland fires + decay.
[10] $CO_2$ emissions from organic soils. Tier 1 approach. EF=20 $tC.ha^{-1}.yr^{-1}$ (IPCC 2006). Only for the six crop types reported by the agricultural
soils (maize, soya, sorghum, wheat, barley, millet). $N_2O$ emissions not included.
[11] $CO_2$ emissions from organic soils. Tier 1 approach. EF=20 $tC.ha^{-1}.yr^{-1}$ (IPCC 2006). $N_2O$ emissions not included.
[12] $CO_2$ for fuelwood is part of the energy balance.
[13] CH4 and N2O emissions for peat, forests and woodland, savannahs and agriculture fires.
[14] CH4, N2O emissions from fire in humid tropical forests and other forests, as well as CH4, N2O from the combustion of organic soils.
[15] CH4, N2O for forest fires + wetland/peatland fires and decay (5A, and 5D classes).
[16] Direct agricultural emissions only
[17] Fertilizers, manure, crop residues
[18] Synthetic fertilizers + Manure applied to soils + Crop residues + Manure applied to pastures.
[19] Indirect emissions



|  | Hotspots | FAOSTAT | EDGAR | Houghton | Baccini | EPA | AR5 |
|---|---|---|---|---|---|---|---|
| Deforestation |  |  | ▓ |  |  |  |  |
| Fire |  | ▓ |  |  | ▓ |  | ▓ |
| Wood-harvesting |  |  | ▓ |  |  |  |  |
| Livestock |  |  |  |  |  |  |  |
| Cropland | ▓ |  |  |  |  | ▓ |  |
| Paddy Rice |  |  |  |  |  |  |  |
| Peatland | ▓ |  |  |  |  | ▓ |  |
|  |  |  |  |  |  |  |  |
| Other |  |  |  | Forest sinks |  |  | Forest sinks |

**Table 4:** summary of the least reliable emission sources (dark grey) for the analysed datasets in this study.

