# Peer review of "Multi-gas and multi-source comparisons of six land use emission datasets"

_Biogeosciences, 2016_

## Referee Comment (RC1) · Anonymous Referee #1 · 2 Aug 2016

General comments:

This manuscript compared the gross AFOLU emissions from 6 land use emission datasets over tropical regions, including their newly published 'Hotspots' dataset combined from multi-studies (Companion Paper bg-2016-99). Authors well identified the agreements/differences between datasets not only by aggregated emissions, but also by emission sources, by gases, and by regions/countries. It further revealed the efforts to make in the future assessments on AFOLU emissions. In my view, this kind of study could really help people understanding the strengthens/weakness of available land use

emission datasets before using them. However, there are several issues should be addressed.

Specific comments:

Table 2a is missing, which makes hard to understand part of the results. Table 3 is really not easy to understand even with the explanations in the main text. It is necessary to think about rearranging it in a clearer way. For example, gas-specific sub-tables might be used, and sources could be list in the table instead of using numbers. Sect. 3.2.2: Forest degradation is not shown in the figures / tables (e.g., indicated with a '{' as the sum of fire and wood harvesting). But suddenly as a parallel section as wood harvesting and fire, it might confuse readers. l. 570: Please further justify the 'least reliable' emission sources for each dataset? For example, which criteria(s) the assessment is(are) based on? l. 594-612: The discussion on RF is not the objective of this manuscript, which appears to unnecessary given the already long main text.

Technical corrections:

l. 216-218: Duplication of l.211-212. l. 323: It might be better to replace the 'our' by the name of dataset, since the objective of this manuscript is comparison rather than presenting a dataset. l. 341: 2000-2009 is indicated for the value of 4.03 PgCO2 yr-1. l. 474: ';' is redundant or maybe there are more reference?

---

## Referee Comment (RC2) · Anonymous Referee #2 · 18 Aug 2016

The main focus of this paper is to evaluate differences in estimates of emissions of greenhouse gases from Agriculture, Forestry, and other Land Use among a suite of data sets. The authors set out to explain the data sources, the intent (and scope) of the sources. This leads then to a nice description of the differences in total greenhouse gas emissions integrated over the tropics, regionalized emissions (across continents and for different country), analysis by source type as well as analysis among different greenhouse gas species (CO2, N2O, and CH4). Overall I thought this is a great contribution and helps readers to 'navigate' the different data sets. The following concerns are minor, but I hope they ultimately will help the reader to better understand

your analysis and your conclusions.

Estimations of country level emissions: this is not clear to me. For the 3 data sets this would results into a sample of n=3 for which you calculated the coefficient of variation. Did you then calculate the percentiles of the coefficients? Does this not imply a false sense of agreement/disagreement? Lets say country 1 results into emissions of 20, 21, and 19, while country 2 has emissions of 2,3 and 1. The coefficient of variation leads to much higher uncertainty in country 2, although the absolute emissions are exactly the same. Perhaps the author could discuss the possibility of other metrics such as the variability of per area emissions (per country) among data sets.

I have trouble to find where degradation fits in. It is not included in any of the graphs or tables, yet the authors spend a lot of time describing it in the methods. In other places it is put in the same bucket as fire and wood harvest. I suggest to refine either the result or the method section to put degradation into the correct context. Similarly, the figures show data for deforestation, and although this is intuitive to many readers, I think a good definition (and how it is being used in context with the data set and this analysis) is important.

Table 5 is not referenced in the text. But it seems an important table. A paragraph in the results/discusson or in the conclusion could really help summarizing in which category the datasets excel and where they are less reliable.

Minor comments and editorial suggestions

L37: Suggest "anthropogenic <greenhouse> gas emissions"

L39: "Global comparison. . ." This is a somewhat awkward sentence – rephrase

L41: suggest i.e. instead of e.g.

L52: instead of paranthesis you may use "with fire leading the difference"

L55: How much of the disagreement stems from incompleteness of the data

L58: I am not a big fan of using etc. . . - but this may be a personal opinion

L65: suggest "Modelling studies suggest that  to keep . . ."

L74: Reading the Anderson, 2015 text, I am not sure whether Anderson made that claim (while he is sceptic about "optimism" in fossil fuel mitigation strategy - suggest reformulation.

L80: This may be the decision also for copy editing, but I think the abbreviation should be preceded by the full Agriculture, Forestry, and other Land Use, although it is explained in the abstract.

L80: unit PgCO2.e.yr-1: I am wondering whether the e for the equivalent should be clarified.

L81: Abbreviation GHG needs to be properly introduced.

L115: Is this PgC or PgCO2?

L119: The statement starting with "These datasets . . ." could benefit with a reference.

L138: I suggest to mention here why the focus in on tropics, instead of burying the rationale in the methods.

L142: In the beginning: Delete the lonely ")"

L149: The discussion about source and sink, net vs. gross can be tightened here. It appears that several statements are repeated.

L177: I think it may be worthwile to briefly (a couple of sentences) explain what the tiers are.

L183: I suggest to use "changes in biomass" and "changes in soil carbon" to highlight that the datasets report the deltas.

L216: The sentence starting with "Unlike other" is a repetition – check L 211

L261: "some of the datasets used", please specify all the datasets that derive their emissions from remote sensing

L271: "To facilitate..." I have a hard time understanding this sentence – possible to rephrase?

L386: Please define CWD abbreviation (or just use coarse woody debris since it is only used once)

L566: Use <change> in SOC, also is the abbreviation properly explained?

L580: What are the units for the numbers?

L587: It is not clear what the FAOSTAT omissions are

L589: try to rephrase "excluding CO2 from aboveground biomass". – "FAOSTAT does not include CO2 emissions from burned biomass" – is this FAOSTAT assumes that fire frequencies are constant through time, and thus the CO2 budget remains unaffected?

L618: "In detriment to sectorial comparisons" – is this a reference to analysis presented in this manuscript?

L629: Is it possible that the good agreement in places where emissions peak is an artefact of the analysis, since relative errors are used (see also my comment above)?

L640: I suggest to use "added" instead of "coming"

L640: Doesn't the A in FRA is assessment? – suggest to delete assessments

L667: Missing reference

L706: direct data on forest degradation is missing (see also my comment above)

L708: Isn't the lifetime of CO2 included in the CO2 equivalent calculation?

L712: What is meant with variability? Also the use of "most" may not be appropriate since there are only two non-CO2 greenhouse gases. Overall, I think this bullet point

should be rephrased

L717: I guess the authors mean that differences among the data sets are as big (or bigger) than the differences among sectors/categories

Figure 1: Why is there suddenly a reference to EDGAR JRC, while in the main text it is referred only to EDGAR. The figure also offers to explain the reader a bit more about the peculiarities if the data. Hotspot is the only data set that has error estimate. EPA has no FOLU emissions calculated while Houghton has not calculated Ag emissions.

Figure 2: why is the Baccini data included here, but not in figure 1 or 3?

Figure 4 caption: typo "peatland"

Figure 1-3 why are the AR5 data not included – I know they are gleaned from the report's figure, but they could stack up against your summary data?

Table 2a is missing

Table 3: This seems to be an important table, but highly cryptic. I suggest to use the acronym of the datasets instead of the numbers

Table 4: "Other" is really only Forest Sinks – so perhaps use "Forest Sinks" as category.

---

## Author Response (AR1)

**Comments to reviewers.  Manuscript bg-2016-244**

We would like to thank the 2 anonymous reviewers for their useful and constructive comments. They have helped us improve our manuscript.

Please note that reference to Lines in our responses correspond to the track-change free manuscript.

Some general comments before entering starting with the detailed responses to the reviewers:

1. The title of the manuscript has been modified to clarify its focusing on the tropics: '**Multi-gas and multi-source comparisons of six land use emission datasets and AFOLU estimates in the Fifth Assessment Report, for the tropics for 2000-2005**'.
2. The affiliation of the first author has been modified to match its latest update.
3. A last author has been included, which was erroneously missing.
4. We would kindly request to include this manuscript, if finally accepted, under the special issue: **'Hotspots of greenhouse emissions from terrestrial ecosystems on global and regional scales'**

**Referee 1**

1. *Table 2a is missing, which makes hard to understand part of the results:* Sorry about this mistake. The full table is now inserted.
2. *Table 3 is really not easy to understand even with the explanations in the main text. It is necessary to think about rearranging it in a clearer way. For example, gas-specific sub-tables might be used, and sources could be list in the table instead of using numbers.* We agree with the reviewer that this table is complex. We have changed the numbers by the name of the datasets and have clarified in the table caption that degradation is formed by wood harvesting and fire. We have, however, kept the superindices because when we tried to reformat it as text, the final tables per gas were enormous
3. *Sect. 3.2.2: Forest degradation is not shown in the figures / tables (e.g., indicated with a '{' as the sum of fire and wood harvesting). But suddenly as a parallel section as wood harvesting and fire, it might confuse readers.* Yes, we see the reviewer;s point. We believe that some definition of forest degradation was useful, as an introductory section to the emissions that lead to forest degradation in our research: fire and wood harvesting. To avoid confusions we have inserted ':wood harvesting and fire emissions' in the caption of section 3.2.2, and have changed the captions of Fire emissions and wood harvesting to 3.2.2.1 and 3.2.2.2. We have changed the remaining captions of this section accordingly.
4. *l. 570: Please further justify the 'least reliable' emission sources for each dataset? For example, which criteria(s) the assessment is(are) based on?* The reviewer is right here. We have explained better what 'least reliable' means to us, which should be understood after reading the differences among databases in section 3.2 (line 575-577)
5. *l. 594-612: The discussion on RF is not the objective of this manuscript, which appears to unnecessary given the already long main text.* Agreed, shortened (l.603-613)

   **Technical corrections:**
6. *l. 216-218: Duplication of l.211-212:* Agreed, removed.
7. *l. 323: It might be better to replace the 'our' by the name of dataset, since the objective of this manuscript is comparison rather than presenting a dataset.* Agreed, we have removed 'our' from the text and substitute it by 'the Hotspots database'.
8. *l. 341: 2000-2009 is indicated for the value of 4.03 PgCO2 yr-1.* We have eliminated the year 2010 and changed it for the period 2000-2009 which is the way how it appears in Figure Fig. 11.2 in Chapter 11 of WGIII, IPCC AR5. Source: https://www.ipcc.ch/pdf/assessment-report/ar5/wg3/ipcc_wg3_ar5_chapter11.pdf. (line 345).
9. *l. 474: ';' is redundant or maybe there are more reference?* Corrected.

**Referee 2**

1. ***Estimations of country level emissions: this is not clear to me. For the 3 data sets this would results into a sample of n=3 for which you calculated the coefficient of variation. Did you then calculate the percentiles of the coefficients? Does this not imply a false sense of agreement/disagreement? Lets say country 1 results into emissions of 20, 21, and 19, while country 2 has emissions of 2,3 and 1. The coefficient of variation leads to much higher uncertainty in country 2, although the absolute emissions are exactly the same. Perhaps the author could discuss the possibility of other metrics such as the variability of per area emissions (per country) among data sets.*** This comment is very useful because we had not noticed it. There is indeed a methodological bias that makes countries with smaller emissions show larger variability (lower agreement among databases) than countries with higher emissions. Offering emission intensities (area rated emissions) would respond to a different question and we have preferred to exclude it. We have re-estimated variability among datasets at the country level using standard deviations, and have included a section in the Supplement, to discuss the differences between statistical choices to contrast data dispersion. There we briefly explain the use of three statistics: coefficient of variation, standard deviation and adjusted standard deviation - considering a correction factor that accounts for a country's contribution to the tropical emission budget-. We include here some example pictures that visually present these differences.

2.
[Figure]

3.

[Figure]

Figure 1: Country emission variability for AFOLU emissions, for the Hotspots, FAOSTAT and EDGAR datasets, using the coefficient of variation or standard deviations as statistics for data dispersion.

[Figure]

Figure 2: Country emission variability for AFOLU emissions; for the Hotspots, FAOSTAT and EDGAR datasets, using standard deviations or an adjusted standard deviation as statistics for data dispersion.

4. *I have trouble to find where degradation fits in. It is not included in any of the graphs or tables, yet the authors spend a lot of time describing it in the methods. In other places it is put in the same bucket as fire and wood harvest. I suggest to refine either the result or the method section to put degradation into the correct context. Similarly, the figures show data for deforestation, and although this is intuitive to many readers, I think a good definition (and how it is being used in context with the data set and this analysis) is important*. Degradation is defined in lines 399-407. We have addressed this claim by re-arranging the captions of fire and wood harvesting in the results/discussion (lines 398, 427, 486) so that they are part of the section of degradation. We have also reminded readers in the graphs, and table captions that degradation in this comparative assessment would be the sum of fire and wood harvesting emissions. We have chosen, however, to retain the degradation section in the results, since we believe that the description of what is degradation in forests and how the datasets are including it, or not, by means of fire and wood harvesting emissions (and other excluded sources) offers interesting information to the readers. Deforestation is explained in the results-discussion and it is not defined as a single concept since different datasets define it differently (see lines 374-376)

5. *Table 5 is not referenced in the text. But it seems an important table. A paragraph in the results/discusson or in the conclusion could really help summarizing in which category the datasets excel and where they are less reliable*: Please note that there is no table 5. Table 4 is referred in lines 575 in the text. We subjectively suggest which are the least reliable sources of emissions (see comment 3, Referee 1) but we do not include best performing emission sources because they are difficult to identify. Thus, methods can differ but be correct.

**Minor comments and editorial suggestions**

*L37: Suggest "anthropogenic gas emissions".* Sorry, the abstract is word limited. Not included.

*L39: "Global comparison. . ." This is a somewhat awkward sentence – rephrase.* Thank you, changed to 'comparisons of global AFOLU emissions'...

*L41: suggest i.e. instead of e.g.* Done

*L52: instead of paranthesis you may use "with fire leading the difference".* Done

*L55: How much of the disagreement stems from incompleteness of the data C2.* We dont follow well this comment. Have not acted upon it.

*L65: suggest "Modelling studies suggest that to keep . . ."* Already written as so in the original version.

*L74: Reading the Anderson, 2015 text, I am not sure whether Anderson made that claim (while he is sceptic about "optimism" in fossil fuel mitigation strategy - suggest reformulation.* Yes, we see the reviewer's point. The reference has been eliminated in L74.

*L80: This may be the decision also for copy editing, but I think the abbreviation should be preceded by the full Agriculture, Forestry, and other Land Use, although it is explained in the abstract.* Done, l79

*L80: unit PgCO2.e.yr-1: I am wondering whether the e for the equivalent should be clarified.* We believe there is no clarification needed. It is accepted as an standing alone 'e', or so has it been in our previous publication in the same journal.

*L81: Abbreviation GHG needs to be properly introduced.* Done

*L115: Is this PgC or PgCO2?* It was correctly written as PgC and we have included the PgCO2e estimate to compare.

*L119: The statement starting with "These datasets . . ." could benefit with a reference.* Agreed, included in L120 and 122.

*L138: I suggest to mention here why the focus in on tropics, instead of burying the rationale in the methods.* This is explained a bit later, and would rather keep it where it is. Please go to the study area section (lines 149-153)

*L142: In the beginning: Delete the lonely ")"* Done

*L149: The discussion about source and sink, net vs. gross can be tightened here. It appears that several statements are repeated.* Agreed. This topic was extensively debated in our accompanying paper so we have now reshaped/shortened this paragraph (l 157-167) and referred the readers to Roman-Cuesta et al. 2016.

*L172: I think it may be worthwile to briefly (a couple of sentences) explain what the tiers are.* I agree with the reviewer, but since the paper is already very long, we have included a definition in the Supplement and referred the readers there.

*L216: The sentence starting with "Unlike other" is a repetition – check L 211* Thank you. Eliminated

*L261: "some of the datasets used", please specify all the datasets that derive their emissions from remote sensing* Only deforestation emissions fully rely on remote sensing. Other emissions use remote sensing (fire, wood harvesting, agriculture...but apply emission models. We have changed the sentence (l. 258)

*L271: "To facilitate. . ." I have a hard time understanding this sentence – possible to rephrase?* Paragraph has been rewritten (l. 270-275)

*L386: Please define CWD abbreviation (or just use coarse woody debris since it is only used once)* Done, l.389

*L566: Use in SOC, also is the abbreviation properly explained?* Done, L. 570

*L580: What are the units for the numbers?* We have changed paragraph. It was hard to understand as it was written.

*L587: It is not clear what the FAOSTAT omissions are.* We have referred readers to section 3.2.2.1 (fire differences among datasets) to clarify this point. L. 427-449

*L589: try to rephrase "excluding CO2 from aboveground biomass". – "FAOSTAT does not include CO2 emissions from burned biomass" – is this FAOSTAT assumes that fire*

*frequencies are constant through time, and thus the CO2 budget remains unaffected?* No, FAOSTAT fully excludes aboveground $CO_2$ fire emissions, to place them instead in net deforestation. Please read fire section, lines 433-435. We have clarified this section a bit.

*L618: "In detriment to sectorial comparisons" – is this a reference to analysis presented in this manuscript?* We have rewritten the material section for the country comparisons to make it more comprehensible. See section 2.5 Country emissions. Line 289-298. We have also redone Sect. 3.4 Lines 615-620

*L640: I suggest to use "added" instead of "coming":* done

*L640: Doesn't the A in FRA is assessment? – suggest to delete assessments:* this has been clarified. Assessments referred to the different FRAs (1990,2000,2005, 2010, etc) since FAOSTAT gets updated with each new FRA.

**L667: Missing reference:** Corrected. Lin. 656

**L706: direct data on forest degradation is missing (see also my comment above):** we do not use direct data on degradation since there are no degradation emission datasets spatially explicit, to our knowledge. Instead, we add the emissions from fire and wood harvesting and consider them to be our forest degradation emissions. See comment 4 of Reviewer 2.

*L708: Isn't the lifetime of CO2 included in the CO2 equivalent calculation?* Only warming potentials are considered in the transformation to equivalents. However, this section refers to the uncertainty of $CO_2$ emissions not to emission estimates.

*L712: What is meant with variability? Also the use of "most" may not be appropriate since there are only two non-CO2 greenhouse gases. Overall, I think this bullet point C4 should be rephrased:* Yes, we understand the reviewer's confusion. We have improved the conclusions of the gases ($CO_2$ and $N_2O$) by referring to the correspondent figures. L. 733-740

*L717: I guess the authors mean that differences among the data sets are as big (or bigger) than the differences among sectors/categories.* Rewritten. Lines 743-749

*Figure 1: Why is there suddenly a reference to EDGAR JRC, while in the main text it is referred only to EDGAR. The figure also offers to explain the reader a bit more about the peculiarities if the data. Hotspot is the only data set that has error estimate. EPA has no FOLU emissions calculated while Houghton has not calculated Ag emissions.* Agreed, done.

*Figure 2: why is the Baccini data included here, but not in figure 1 or 3?* The reviewer is right. We have redone all pictures. Figure 1 was wrongly missing Baccini's data. We have now corrected it. Baccini's data are, however, not offered in a disaggregated spatial manner so they can only be part of figures at tropical scale (i.e. Fig 1 and 2, but not 3).

*Figure 4 caption: typo "peatland"* Thanks, corrected

*Figure 1-3 why are the AR5 data not included – I know they are gleaned from the report's figure, but they could stack up against your summary data?* Yes, the reviewer's got a point here but we originally decided to exclude the AR5 data in the graphs because they require too many explanations (i.e. net emissions instead of gross but sinks are only partial since they do not include forest sinks of standing forests if not disturbed or undergoing shifting cultivation recovery), data are for a different period (2000-2009) and we do not have spatially disaggregated data from where we could exactly extract our tropical study area. Some tropical data are offered in graphs within the AR5 Chapter 11, so the data can be more or less derived for a text discussion, but not so good for a numeric comparison.

*Table 2a is missing:* Corrected.

*Table 3: This seems to be an important table, but highly cryptic. I suggest to use the acronym of the datasets instead of the numbers.* Yes, the table is still complex, but we improved the caption to expand its comprehension and changed the numbers by the datasets.

*Table 4: "Other" is really only Forest Sinks – so perhaps use "Forest Sinks" as category.* Done

[revised manuscript text omitted]